

# Aerosol Optical Properties at SORPES in Nanjing, East China

Yicheng Shen[1], Aki Virkkula[1,2,3,4,*], Aijun Ding[1,2*], Jiaping Wang[1], Xuguang Chi[1,2],

Wei Nie[1,2], Ximeng Qi[1,2], Xin Huang[1,2], Qiang Liu[1], Longfei Zheng[1,2], Zheng Xu[1,2],

Tuukka Petäjä[4], Pasi P. Aalto[4], Congbin Fu[1,2], and Markku Kulmala[4]

[1]Joint International Research Laboratory of Atmospheric and Earth System Sciences, and School of Atmospheric Sciences, Nanjing University, 210023, China

[2]Collaborative Innovation Center of Climate Change, Jiangsu Province, China

[3]Finnish Meteorological Institute, FI-00560, Helsinki, Finland

[4]Department of Physics, University of Helsinki, FI-00014, Helsinki, Finland

* Correspondence to A. Ding (dingaj@nju.edu.cn) or A. Virkkula (Aki.Virkkula@fmi.fi)

**Abstract**

Aerosol optical properties (AOPs) and supporting parameters – particle number size distributions, mass concentrations and trace gases ($NO_x$ and $NO_y$) – were measured at SORPES, a regional background station in Nanjing, China from June 2013 to May 2015. The aerosol was highly scattering: the average scattering coefficient was $\sigma_{sp} = 410 \pm 320$ $Mm^{-1}$, the absorption coefficient $\sigma_{ap} = 26 \pm 19$ $Mm^{-1}$ and the single-scattering albedo SSA = 0.93 ± 0.03 for the green light. The SSA in Nanjing appears to be slightly higher than published values from several other sites in China and elsewhere. The average Ångström exponent of absorption (AAE) for the wavelength range 370 – 950 nm was 1.04 and the AAE range 0.7 – 1.4. These AAE values can be explained with different amounts of non-absorbing coating on pure BC cores and different core sizes so the data do not suggest any significant contribution to absorption by brown carbon. The AOPs had typical seasonal cycles with high $\sigma_{sp}$ and $\sigma_{ap}$ in winter and lower in summer: the averages were $\sigma_{sp} = 545 \pm 425$ $Mm^{-1}$ and $\sigma_{ap} = 36 \pm 24$ $Mm^{-1}$ in winter and $\sigma_{sp} = 364 \pm 294$ $Mm^{-1}$ and $\sigma_{ap} = 20 \pm 13$ $Mm^{-1}$ in summer. The intensive AOPs had no clear seasonal cycles, the variations of them were rather related to the evolution of pollution episodes. The diurnal cycles of the intensive AOPs were clear and in in agreement with the cycle of the particle number size distribution. The diurnal cycle of SSA was similar to that of the air photochemical age, suggesting that the darkest aerosol originated from fresh traffic emissions. A Lagrangian retroplume analysis showed that the sources of high $\sigma_{sp}$ and $\sigma_{ap}$ are mainly in eastern China. Synoptic weather dominated the cycle of AOPs in a temporal scale of 2-7 days. During pollution episodes, modeled PBLH decreased, whereas $PM_{2.5}$ concentrations, $\sigma_{sp}$ and $\sigma_{ap}$ typically increased gradually and remained high during several days but decreased faster, sometimes by even more than an order of magnitude within some hours. During the growth phase of the pollution episodes the intensive AOPs evolved clearly. The mass scattering efficiency MSE of of $PM_{2.5}$ grew during the extended pollution episodes from ~4 $m^2$ $g^{-1}$ to ~6 $m^2$ $g^{-1}$ and the mass fraction of $BC_e$ decreased from ~10% to ~2 % during the growth phase of the episodes. Particle growth resulted in $b$ decreasing from more than 0.16 to less than 0.10, SSA growing from less than 0.9 to more than 0.95 and radiative forcing efficiency RFE growing from less than -26



W m$^{-2}$ τ$^{-1}$ to more than -24 W m$^{-2}$ τ$^{-1}$. In other words, the darker aerosol – the aerosol that had a higher BC mass fraction – had a more negative radiative forcing efficiency, i.e., they have the property of cooling the atmosphere more efficiently per unit optical depth than the aerosol with the higher SSA and a lower BC mass fraction. This counterintuitive result is due to the size of the particles: the upscatter fraction of small particles is higher than that of the big ones which more than

compensates the darkness of them. The RFE probability distribution at SORPES was clearly more narrow than at a clean background site which is in agreement with a published RFE climatology.

## 1. Introduction

Atmospheric aerosols alter the radiation budget of earth system directly by scattering and absorbing solar radiation and

indirectly by acting as cloud condensation nuclei (CCN), thus affecting cloud formation, cloud optical properties and cloud lifetime (IPCC, 2013). Radiation forcing of aerosol exerts significant impacts on the climate system and contributes the largest uncertainty in the assessment of climate change both regionally and globally (IPCC, 2013). Such uncertainty is not only due to highly inhomogeneous spatial distributions as well as temporal variations of aerosol, but also due to limited measurement of aerosol chemical composition and size distributions which aerosol optical properties (AOPs)

depend on. Black carbon (BC) aerosols, or light-absorbing carbon (LAC) aerosols are especially important due to their strong capability of light absorption. It has been estimated that the radiative and climate impacts of BC is the second strongest contributor to global warming (Ramanathan et al., 2007) and can also influence rainfall, large-scale circulation and hydrological cycles (Menon et al., 2002). Recent studies also indicate that the absorbing aerosols heat the air and changes the vertical temperature profile thus have an influence on PBL structure (e.g., Ding et al., 2013a; Ding et al.

2016a; Wilcox et al., 2017). Light scattering aerosol over polluted continental areas consist mainly of secondary aerosol produced by heterogeneous reactions, which greatly increase the aerosol scattering coefficient and play an important role in haze events in China (Huang et al., 2014).

During recent decades, many comprehensive studies of aerosol optical properties have been conducted in China,

especially in the three large urban agglomerations the North-China-plain (NCP), the Pearl River Delta (PRD), and the Yangtze River Delta (YRD) region (Kim et al., 2004; Andreae et al., 2008; Yan et al., 2008; Cheng et al., 2008; Yang et al., 2009; Ma et al., 2011; Zhuang et al., 2014, 2016; Wang et al., 2017). These three regions are highly industrialized with an extremely high population density; due to the complex anthropogenic emissions and consequent chemical reactions these three regions not only have some of the highest aerosol loadings worldwide but also some of the greatest

uncertainties of radiative forcing of aerosols. The fast urbanization and industrialization process along with new technology and policy make the emission characteristics change year by year.

To reduce the uncertainties of aerosol climatic effects, long-term continuous measurements of aerosol optical properties, particle size distributions and other relevant parameters including trace gas concentrations and meteorological parameters

have been conducted at the Station for Observing Regional Processes of the Earth System (SORPES), a regional background station in the YRD (Ding et al, 2013b; Ding et al., 2016b). SORPES is located in the northwestern part of



YRD area and to the northeast of Nanjing urban area. The complex monsoon and synoptic weather together with the complex emission source surrounded may have important impacts on aerosols in this region (Ding et al., 2013b).On the other hand, the aerosol optical properties influence radiative transfer and further modify meteorological processes, such as the planetary boundary layer (PBL) and weather (Ding et al., 2013a). Ding et al. (2016a) and Petäjä et al. (2016) studied

the interactions of pollutants and the PBL and found that high PM and especially BC concentrations enhance the stability of a polluted boundary layer by modifying the PBL temperature profile and by decreasing the surface heat flux. In these papers several month of observational data from SORPES were used, however, there is a lack of analysis on the long-term data and no detailed analysis of how particle size distributions and AOPs evolve during extreme haze pollution episodes. Part of the AOP data measured at SORPES were earlier used for a technical analysis of the interpretation of

absorption photometer data (Virkkula et al., 2015). In this work, we will present two years of AOP observations at the SORPES station and a comprehensive analysis of their temporal variations and relationships with particle size distributions and transport.

## 2. Measurements and methods

### 2.1 Site description

The measurements were conducted at the SORPES station (118°57′10″ E, 32°07′14″N, ~40 m above sea level) from July 2013 to May 2015. The station is located on the top a small hill, ~30 meters above ground level, inside Nanjing University Xianlin Campus, 20 km northeast of downtown Nanjing (Ding et al., 2013b; Ding et al., 2016b; Xie et al., 2015). SORPES is surrounded by several campuses and residential areas and there is no industry within 3 km from the station. A new

highway ~1 km to the west of SORPES was opened to traffic in 2014. The prevailing easterly wind, i.e. northeast in winter and southeast in summer (Ding et al., 2013b), minimizes the effect from Nanjing downtown and from the highway. With some consideration on the data analysis, the site can still be considered as a regional background station with little local influence (Ding et al., 2016b).

### 2.2 Measurements

Aerosols and trace gases were measured at SORPES inside a two-floor building on the hillside close to the top of the hill. The roof of this building is approximately at the same level as the flat hilltop, which has a flat area of over 2500 m$^2$ with standard meteorological observations (wind, temperature, relative humidity, solar radiation) and a 75-m tall mast for flux measurements. There are several inlets among the roof and most instruments inside the building have their individual

inlets. All meteorological parameters and radiation were obtained from the standard meteorology observation field.

### 2.2.1 Aerosol optical measurements

The scattering and backscattering coefficients ($\sigma_{sp}$ and $\sigma_{bsp}$, respectively) were measured at three wavelengths ($\lambda$ = 450 nm, 525 nm and 635 nm) using an integrating nephelometer (Aurora 3000, Ecotech). The sample air was taken through a

2-meter stainless tube, the top of which is 1 meter above the roof. The inlet has a rain cap and an external heater to prevent condensation. The zero of the nephelometer was checked by filtered air every day and the span by Tetrafluoroethane



(R134a) every two weeks. The monthly average RH at SORPES varies from 65% to 80% (Ding et al., 2013b) and the aerosol hygroscopic growth is usually significant when RH increases above 50% (Zhang et al. 2015; WMO, 2016). To reduce the RH, we use an internal heater so that the RH of the sample air is below 50% for most of the time. Since the internal heater of the nephelometer often malfunctioned, ~25 - 30% percent of data suffered from high RH. The respective

data were corrected as will be discussed later on. Light absorption was measured using a 7-wavelength aethalometer (AE-31, Magee Scientific) at λ=370 nm, 470 nm, 520 nm, 590 nm, 660 nm, 880 nm, and 950 nm. The aethalometer is a filter-based instrument that measures light attenuation from which light absorption can be calculated. The detailed calculation will be discussed below. The aethalometer shares the same $PM_{2.5}$ cyclone inlet with several trace gas analyzers. The sample air is taken through a stainless steel tube to the instruments. The flowrate for the aethalometer was set to 5.0 LPM

for the whole period. An internal flowmeter records the real-time flowrate continuously, and flow checks were conducted twice a year using a bubble flowmeter (Gilibrator system, Gilian). The time resolution of the aethalometer data was set to 5 minutes and it was set to change the sampling spot when the maximum attenuation was 125. These settings were used for the whole period.

**2.2.2 Supporting measurements**

Particle number size distributions were measured using a custom-made Differential Mobility Particle Sizer (DMPS) in the size range of 6 – 800 nm (mobility diameter) and an Aerodynamic Particle Sizer (APS, TSI Model 3321) in the size range of 0.52 – 20 µm (aerodynamic diameter). More details can be seen in Qi et al. (2015). The number size distributions were used here for modeling scattering coefficients, for calculating effective diameters and estimating particle mass

concentrations as will be discussed below. Unfortunately the APS was operational for a few months only so all the size distribution related parameters below were calculated from the DMPS data.

Mass concentrations of particles smaller than 2.5 µm ($PM_{2.5}$) were measured with an online analyzer based on the light scattering and beta ray absorption (Thermo Scientific, 5030 SHARP, USA). The trace gas measurements ($NO_x$ and $NO_y$) used in this work were conducted with a $NO-NO_2-NO_x$ Analyzer (Model 42i, Thermo Scientific, USA) and a NO-DIF-

$NO_y$ Analyzer (Model 42i-Y, Thermo Scientific, USA). These data were used for estimating the photochemical age of air masses. More details for $PM_{2.5}$, trace gases and meteorological parameters can be found in Ding et al. (2013b and 2016b).

**2.3 Data processing**

**2.3.1 Corrections for scattering coefficient**

First the truncation error of the scattering measurements was corrected according to Müller etal. (2011). In addition, since 28.6% of total data were measured when the sample air relative humidity $RH_{sample}>50$ % due to an intermittent fault of the internal heater of the nephelometer, the scattering coefficients were corrected for hygroscopic growth in order to maximize the data availability when the internal heater of the nephelometer was malfunctioning. Hygroscopic aerosols

take up water as humidity increases thus increasing $\sigma_{sp}$. The impact of relative humidity on $\sigma_{sp}$ is defined as the scattering enhancement factor $f$ (RH, $\lambda$):





$$f(\text{RH}, \lambda) = \sigma_{sp}(\text{RH}, \lambda) / \sigma_{sp}(\text{dry}, \lambda) \tag{1}$$

where $\sigma_{sp}(\text{dry}, \lambda)$ and $\sigma_{sp}(\text{RH}, \lambda)$ represent scattering coefficients at wavelength $\lambda$ in dry and humid conditions, respectively. We used the parameterization with the equation

$$f(\text{RH}) = c \cdot (1-\text{RH})^{-g} \tag{2}$$

in this study. The constants $c$ and $g$ were set to 0.85 and 0.29, respectively, according to Zhang et al. (2015) who derived them from measurements in Lin'an, another regional background site inYRD, ~200km southeast from our site. All scattering coefficients measured with sample relative humidity $\text{RH}_{sample} > 50$ % were corrected according to (2) to RH=50 %. By considering that the RH sensor inside the nephelometer may not be as accurate as the one at the meteorological station, we recalculated the sample RH using the Clausius–Clapeyron equation. It was assumed that the

absolute humidity of the sample air and at the RH sensor 20m away from the inlet are the same. The pressure and temperature of the nephelometer were used for correcting the scattering coefficients to Standard Temperature and Pressure (STP) condition (T = 273.15 K, p = 1013 hPa). All aerosol optical properties discussed in this paper are use STP condition unless otherwise specified.

**2.3.2 Absorption coefficient**

The aethalometer does not measure absorption coefficient directly, instead, it measures the light attenuation (ATN) of aerosol-loaded spots on quartz filters. The attenuation coefficient $\sigma_{atn}$ at wavelength $\lambda$is calculated from

$$\sigma_{atn}(\lambda) = \frac{A\Delta ATN(\lambda)}{Q\Delta t} \tag{3}$$

where Q is the flow rate and A is the spot size and $\Delta$ATN is the change of attenuation during the time step $\Delta$t. The

Aethalometer firmware converts $\sigma_{atn}$ to equivalent black carbon (BC$_e$) (Petzold et al., 2013) mass concentration by dividing it with a wavelength-dependent mass attenuation coefficient of 14625 $m^2 g^{-1}$ nm/ $\lambda$. However, it is not as straightforward to calculate the absorption coefficient $\sigma_{ap}$. Several algorithms for calculating $\sigma_{ap}$ from aethalometer data have been presented, e.g., Weingartner et al. (2003), Arnott et al. (2005), Schmid et al.(2006), Virkkula et al. (2007), and Collaud Coen et al. (2010). In principle they can all be presented in the form of

$$\sigma_{ap} = \frac{f\sigma_{atn} - s\sigma_{sp}}{C_{ref}}, \tag{4a}$$

or

$$\sigma_{ap} = \frac{\sigma_{atn} - s'\sigma_{sp}}{C_{ref}R}, \tag{4b}$$

where $f$ and $R$ are functions that correct for the loading, $s$ and $s'$ the fraction of scattering coefficient that results in a change of ATN and would be interpreted as absorption and BC$_e$ if not taken into account, and C$_{ref}$ is the multiple-scattering

correction factor. There are no unambiguous forms for f, R, s, s', and C$_{ref}$. A recent analysis suggested that f is influenced by the aerosol backscatter fraction (Virkkula et al., 2015). Arnott et al. (2005) and Schmid et al. (2006) suggest that C$_{ref}$ is wavelength-dependent whereas Collaud Coen et al. (2010) used a non-wavelength-dependent C$_{ref}$. In this study, we used the Collaud Coen et al. (2010) algorithm with C$_{ref}$ = 4.26, the value obtained from measurements in a flat region near populated and industrialized areas at Cabauw, the Netherlands (Collaud Coen et al., 2010).



### 2.3.3 Intensive aerosol optical properties

Light absorption and scattering depend on wavelength λ approximately as $\lambda^{-AAE}$ and $\lambda^{-SAE}$, where AAE and SAE are the Ångström exponents of absorption and scattering, respectively. SAE can be calculated from $\sigma_{sp}$ measured at two wavelengths $\lambda_1$ and $\lambda_2$ from:

$$SAE = -\frac{\log(\sigma_{sp,\lambda_1}) - \log(\sigma_{sp,\lambda_2})}{\log(\lambda_1) - \log(\lambda_2)} \tag{5}$$

For multiple wavelengths SAE can also be calculated by taking logarithm of scattering coefficients and the respective wavelengths and SAE is the slope obtained from their linear regression a was done by Virkkula et al. (2011). SAE is typically considered to be associated with the dominating particle size so that large values (SAE > 2) indicate a large contribution of small particles and small values (SAE < 1) a large contribution of large particles. However, this

10   relationship is not quite unambiguous, as discussed by, e.g., Schuster et al. (2006) and Virkkula et al. (2011). In this study, unless otherwise specified, we use $\sigma_{sp}$ at λ = 635 nm and 450 nm to calculate SAE. For the tables also other wavelength pairs were used. To calculate absorption coefficient according to (4), $\sigma_{sp}$ measured at the nephelometer wavelengths were interpolated and extrapolated to the aethalometer wavelengths according to $\sigma_{sp,\lambda x} = \sigma_{sp,\lambda 1} (\lambda_1/\lambda_x)^{SAE}$ .

15   AAE can be calculated from (5) by using $\sigma_{ap}$ instead of $\sigma_{sp}$. AAE is an indicator of the dominant light absorber so that values around 1 indicate absorption by BC (e.g., Bond and Bergstrom, 2006; Bond et al., 2013) and clearly larger values by other absorbers. For example light absorbing organics may yield AAE in the range 3 – 7 (e.g., Kirchstetter and Thatcher, 2012). The interpretation is ambiguous since AAE not only depends on the dominant absorber but also on the size and internal structure of the particles. For instance, for pure BC particles AAE varies may have AAE < 1 and BC particles

20   coated with non-absorbing material may have AAE in the range from < 1 to almost 2 (e.g., Gyawali et al., 2009; Lack and Cappa, 2010). In this study, unless otherwise specified, we use $\sigma_{ap}$ at λ = 370 nm and 950 nm to estimate AAE. For the tables also other wavelength pairs were used.

The ratio of scattering to extinction is the single scattering albedo (SSA)

$$SSA = \frac{\sigma_{sp,\lambda}}{\sigma_{ext,\lambda}} = \frac{\sigma_{sp,\lambda}}{\sigma_{sp,\lambda} + \sigma_{ap,\lambda}} \tag{6}$$

It is a measure of the darkness of aerosols. At low SSA aerosols heat the atmosphere and at high values they cool it, depending also on other parameters (e.g., Haywood and Shine, 1995). SSA is ~0.3 for pure BC particles (e.g., Schnaiter et al., 2003) and 1 for purely scattering aerosol. SSA was calculated for the aethalometer wavelengths. The backscatter fraction

$$b = \frac{\sigma_{bsp}}{\sigma_{sp}} \tag{7}$$

was calculated at λ = 450nm, 525nm and 635nm. It is a measure related to the angular distribution of light scattered by aerosol particles. For very small particles $b$ approaches the value 0.5 (e.g., Wiscombe and Grams, 1976; Horvath et al., 2016) and decreases with increasing particle size. From $b$ it is possible to estimate the average upscatter fraction $\beta$, one





of the properties controlling the aerosol direct radiative forcing (e.g., Andrews et al., 2006). The larger $b$ is, the more aerosols scatter light to space and cool the atmosphere or, heat it less if the aerosol is so dark that it heats the atmosphere as shown in the formula for the top of the atmosphere aerosol radiative forcing efficiency (RFE = $\Delta F/\tau$), i.e., aerosol forcing per unit optical depth ($\tau$):

$$\frac{\Delta F}{\tau} = -DS_0 T_{at}^2 (1 - A_c) \omega_0 \beta \left\{ (1 - R_s)^2 - \left( \frac{2R_s}{\beta} \right) \left[ \left( \frac{1}{\omega_0} \right) - 1 \right] \right\} \tag{8}$$

where $D$ is the fractional day length, $S_o$ is the solar constant, $T_{at}$ is the atmospheric transmission, $A_c$ is the fractional cloud amount, $R_s$ is the surface reflectance, and $\beta$ is the average upscatter fraction calculated from $b$. If the non-aerosol-related factors are kept constant and if it is assumed that β has no zenith angle dependence this formula can be used for assessing the intrinsic radiative forcing efficiency by aerosols (e.g., Sheridan and Ogren, 1999; Delene and Ogren, 2002). The constants used were $D = 0.5$, $S_o = 1370$ W m$^{-2}$, $T_{at} = 0.76$, $A_c = 0.6$, and $R_s = 0.15$ as suggested by Haywood and Shine (1995) and β was calculated from $\beta = 0.0817 + 1.8495b - 2.9682b^2$ (Delene and Ogren, 2002).

### 2.3.4 Properties calculated from the particle number size distributions

The size distributions measured with the DMPS were used to calculate three weighted mean diameters, the geometric mean diameter

$$\text{GMD} = \exp\left( \frac{\sum N_i \ln D_{p,i}}{N_{tot}} \right) \tag{9}$$

the surface mean diameter

$$\text{SMD} = \frac{\sum D_{p,i} S_i}{S_{tot}} = \frac{\sum D_{p,i}^3 N_i}{\sum D_{p,i}^2 N_i} \tag{10}$$

and volume mean diameter

$$\text{VMD} = \frac{\sum D_{p,i} V_i}{V_{tot}}. \tag{11}$$

The mass concentration of particles smaller than 0.8 µm was calculated from $m_{0.8} = \rho_p V_{tot} = \rho_p \sum N_i \frac{\pi}{6} D_{p,i}^3$, where the density of particles $\rho_p$ was assumed to be 1.7 g cm$^{-3}$. The DMPS measures the mobility diameter which for spherical particles equals the physical diameter $D_p$. The aerodynamic diameter of spherical particles with density $\rho_p$ is $D_a = \sqrt{\rho_p / \rho_0} D_p$, where $\rho_0$ is the density of water. For $D_p = 0.8$ µm and $\rho_p = 1.7$ g cm$^{-3}$ this yields $D_a = 1.0$ µm. In the results, therefore, the mass concentration calculated from the number size distributions will be denoted as PM$_1$. The reasoning for the use of density of 1.7 g cm$^{-3}$ was presented by Wang et al. (2017).

The DMPS measurements were continuous throughout the whole study period, while the APS for a few months only. We estimated the contribution of particles smaller than 800 nm to scattering by using the DMPS and APS data measured during one month in July – August 2014. Scattering coefficients were calculated from

$$\sigma_{sp}(\lambda) = \int Q_{sp}(\lambda, D_p, m) \frac{\pi}{4} D_p^2 n(D_p) dD_p \tag{12}$$

where $Q_{sp}$ is the scattering efficiency that depends on particle size ($D_p$), wavelength ($\lambda$) and refractive index ($m$) of the

particles. The results showed that the contribution of particles in the DMPS size range was ~91% of the whole integrated $\sigma_{sp}$. Since the APS data was short we did not model scattering for the whole period, but only discuss the weighted mean diameters from DMPS data.

**2.4 Use of the trace gas data**

The $NO_x$ and $NO_y$ concentrations can be used as a semi-quantitative indicator of the age of air masses since the emission from $NO_x$-emitting sources, mainly road traffic. $NO_x$ is the sum of NO and $NO_2$ and $NO_y$ the sum of $NO_x$ and its oxidation products (e.g. $HNO_3$, PAN and organic nitrates). The photochemical age, denoted as PA below, can be described using the ratio between $NO_x$ and $NO_y$ (Olszyna et al., 1994; Kleinman et al., 2008) and it is usually calculated as the negative logarithm of this ratio, i.e., PA = -log($NO_x/NO_y$). If $NO_x = NO_y$, then -log($NO_x/NO_y$) = 0 and higher when $NO_x$ has had time to be oxidized. In fresh traffic emissions -log($NO_x/NO_y$) is small and high in aged air masses.

**2.5 Modeling**

To estimate the source regions of air masses, backward Lagrangian particle dispersion modeling (LPDM) was conducted by using the Hybrid Single-Particle Lagrangian Integrated Trajectory (HYSPLIT) model developed in the Air Resource Laboratory of the National Oceanic and Atmospheric Administration (Draxler and Hess, 1998; Stein et al., 2015). 48-hour backtrajectories and 72-hour backward retroplumes starting from the surface (i.e., surface residence time of backward simulation for particles released at a specific location) of air-mass on the surface were calculated based on a method developed and evaluated by Ding et al. (2013c). The 48-hour backtrajectories were clustered using HYSPLIT. Description of the clustering method is found at the NOAA ARL web page and will not be described here.

The Planetary Boundary Layer (PBL) height was calculated by WRF modeling using the Yonsei University planetary boundary layer (YSU PBL) scheme. The PBL top is determined using a critical bulk Richardson number of zero, so it is effectively dependent on the buoyancy profile and the PBL top is defined at the maximum entrainment layer (Hong et al., 2006).

**3. Results and discussion**

Below first a general overview of the concentrations and AOPs is given, including a comparison with observed values at other Chinese and foreign sites. Then the seasonal and diurnal variations are discussed. Sources of aerosols are analyzed by using LPDM and back trajectory clusters, wind data and some information on trace gas concentrations. A polluted winter period will be analyzed in detail and finally some relationships between observed and derived AOPs are discussed.

**3.1 Overview of aerosol optical properties**

The daily averaged and range of mass concentration of particles smaller than 2.5 µm ($PM_{2.5}$), scattering coefficients ($\sigma_{sp}$), absorption coefficients ($\sigma_{ap}$), and single-scattering albedo (SSA) of green light during the whole period discussed in the paper are presented in Figure 1. The extensive aerosol properties ($PM_{2.5}$, $\sigma_{sp}$, $\sigma_{ap}$) measured with the three independent





instruments, the mass monitor, the nephelometer and the aethalometer followed each other. They often varied by almost an order of magnitude between consecutive days. There were approximately 2-6 clear pollution episodes a month. In these episodes, PM$_{2.5}$ concentrations and scattering and absorption coefficients typically increased and remained high during several days but decreased faster and remained at the lower level for a shorter period than at the higher level. This will be discussed more below.

The contribution of scattering to total extinction ($\sigma_{ep} = \sigma_{sp} + \sigma_{ap}$) was high enough to keep SSA > 0.9 most of the time. Daily averages with very high SSA > 0.95 were observed throughout the year but especially in summer 2014. On the other hand, in summer 2013 there were several days with SSA < 0.9. A careful comparison of the time series shown in Figure 1 shows that generally SSA was the highest at the peak PM$_{2.5}$ concentration. It will be shown below that this is typically associated with particle growth.

Table 1 shows a statistical overview of the aerosol optical properties and the related supporting parameters PM$_{2.5}$ concentration, GMD and VMD during the period discussed in this paper. The overall data coverage for absorption and scattering coefficients are 91.2% and 95.0% respectively. Since absorption coefficients can only be derived when both nephelometer and aethalometer working properly, the data coverage for absorption coefficient (91.2%) is also the fraction of time when all aerosol optical properties have valid values. The average ± standard deviation of $\sigma_{sp}$ and $\sigma_{ap}$ were 426 ± 327 Mm$^{-1}$ and 26 ± 19 Mm$^{-1}$ at λ = 525 nm and λ = 520 nm, respectively. These two are plotted against each other in Figure 2 together with respective published values observed at selected Chinese and foreign sites presented in Table 2. The results from some foreign sites are included in Fig. 2 to put the SORPES data into a more global context. At most sites $\sigma_{sp}$ and $\sigma_{ap}$ were neither measured with similar instruments nor at the same wavelengths as at SORPES which complicates the comparison. To make the results as comparable as possible, only green wavelength results are shown. The $\sigma_{ap}$ averages were interpolated to the same wavelength at which the green $\sigma_{sp}$ was measured at the same site. In this interpolation, it was assumed that AAE = 1 which undoubtedly increases the uncertainty of the comparison.

The overall average $\sigma_{sp}$ and $\sigma_{ap}$ measured at SORPES were comparable to those at Lin'An in November 1999 and at Pukou (in Nanjing) from March 1 to April 30, 2011. These two sites are both in the YRD area. In Nanjing center, the annual averages were very close to those at SORPES, even though $\sigma_{ap}$ was slightly higher and $\sigma_{sp}$ lower in the center of the city, respectively, which can be explained by the proximity to fresh traffic emissions. Another difference is that Zhuang et al. (2017) used the C$_{ref}$ = 3.56 at λ = 520 nm for calculating $\sigma_{ap}$ from aethalometer data whereas the value C$_{ref}$ = 4.26 was used for all wavelengths in the present study. For comparison, the average $\sigma_{ap}$ at SORPES was calculated also by using the same C$_{ref}$ = 3.56 at λ = 520 nm and plotted against $\sigma_{sp}$ during the above-mentioned periods in Figure 2. When the same C$_{ref}$ is used for processing both sites' data, the average $\sigma_{ap}$ appears to be slightly higher at SORPES than in the center of the city.

The scatter plot reveals also some noteworthy differences between the sites. In the coastal city of Shanghai $\sigma_{sp}$ was lower





and $\sigma_{ap}$ was higher than at the inland sites of YRD (Lin'An and Nanjing) in any of the published results. One possible reason for the difference is that when wind blows from the NE, the prevailing wind direction in YRD, in Shanghai there is less time for the formation and condensation of scattering material on fresh BC particles than at the inland sites. The scattering coefficient at the YRD inland sites is also comparable with published values observed in Beijing and Wuqing,

a site between Beijing and Tianjin in the North China Plain (NCP) and in Guangzhou in the Pearl River Delta(PRD) region in China. On the other hand, $\sigma_{ap}$ is lower at the YRD inland sites than at the NCP and PRD sites which together with high $\sigma_{sp}$ leads to higher SSA than at the other polluted sites. This suggests that there may be differences in the local and regional emissions.

Shangdianzi is a Global Atmosphere Watch (GAW) regional station 100 km northeast of Beijing (Yan et al., 2008). Tongyu is a regional background station located in the semi-arid area of Northeast China even further, about 750 km NE of Beijing (Wu et al., 2012). Both Shangdianzi and Tongyu are partially influenced by highly polluted air from SW and cleaner air from remote areas so the average $\sigma_{ap}$ and $\sigma_{sp}$ levels are clearly lower than in Beijing or YRD. But at both sites the average SSA is slightly lower than at SORPES. The Hong Kong measurements were conducted at the Hok Tsui monitoring station

on the southeast tip of Hong Kong Island facing the South China Sea. Even though it is not far from the city, its aerosols are dominated by sea-salt particles which explains the high SSA (Wang et al., 2017).

In Delhi, India, $\sigma_{ap}$ was the highest, suggesting high BC emissions. However, there also $\sigma_{sp}$ was so high that the SSA was in the same range as at the YRD inland sites. Grenada, an urban site in Spain is an obvious outlier in the comparison,

there the aerosol is the darkest, $\sigma_{ap}$ is at the same level as in Nanjing but $\sigma_{sp}$ clearly lower. Lyamani et al. (2008) suggested that part of the explanation of low SSA in Grenada transport of light-absorbing Saharan dust in Grenada. The lowest $\sigma_{ap}$ and $\sigma_{sp}$ in this comparison were at two continental background sites in the USA, the Amazonian site in Brazil and at the boreal forest site of SMEAR II in Finland. Note that also there SSA was slightly lower than at the YRD inland sites.

The overall SAE average ± standard deviation for the wavelength pair 450 / 635 nm was $1.30 \pm 0.34$ (Table 1) which is very close to the respective value of $1.32 \pm 0.41$ in the center of Nanjing (Zhuang et al., 2017). The overall average backscatter fraction $b = 0.12 \pm 0.02$ at $\lambda = 525$ nm (Table 1) is also very similar to that at several sites around the world (e.g., Delene and Ogren, 2002; Andrews et al., 2011; Virkkula et al., 2011). The average Ångström exponent of absorption (AAE) for the wavelength pair 370 / 950 nm was $1.04 \pm 0.15$ and 98% of the values varied in the range of about 0.7 – 1.4.

Using the wavelength pairs 370 / 880 nm and 470 / 660 nm yielded slightly different AAE distributions (Table 1). The AAE range 0.7 – 1.4 is explainable with different amounts of non-absorbing coating on pure BC cores and different core sizes (Gyawali et al., 2009; Lack and Cappa, 2010) so the data does not suggest any significant contribution to absorption by brown carbon. Note, however, that these values were calculated with the algorithm of Collaud Coen (2010) with a non-wavelength-dependent $C_{ref}$. Algorithms that assume wavelength-dependent $C_{ref}$ (Arnott et al., 2005; Schmid et al.,

2006) would yield higher AAE. For instance, Zhuang et al. (2017) used the latter algorithm and reported AAE = $1.58 \pm 0.23$ for the measuremets in the center of Nanjing.



### 3.2 Seasonal variation of aerosol optical properties

We used the hourly-averaged data measured from June 2013 to May 2015 for the analyses of seasonal cycles of aerosol optical properties and the influencing factors at SORPES. The four seasons are defined as follows: spring: March – May, summer: June – August, autumn: September – November and winter: December – February. The seasonal cycles of 8 parameters: $\sigma_{sp}$, $\sigma_{ap}$, b, SSA, SAE, AAE, $PM_{2.5}$, and GMD are presented in Figure 3 and in Table 3. Both scattering and absorption coefficient have a clear seasonal cycle. In general, they were clearly higher in late autumn and winter than in summer. Both coefficients reached the peak monthly averages in December, $\sigma_{sp}$=618 Mm$^{-1}$ at $\lambda$ = 525nm and $\sigma_{ap}$ =37.7Mm$^{-1}$ at $\lambda$ = 520 nm, more than twice as high as those in August, $\sigma_{sp}$ =256 Mm$^{-1}$ and $\sigma_{ap}$ = 14.4 Mm$^{-1}$. Such a seasonal cycle agrees with the seasonal cycle of $PM_{2.5}$ mass concentrations. Several possible explanations are: 1) in winter the prevailing wind from YRD region or North China Plain (NCP) brings polluted air masses continuously which enhances the pollution while wind blows from different, cleaner directions by the summer monsoon in summer (Ding et al., 2013c demonstrate this point from LPDM simulation); 2) more efficient vertical mixing of the aerosol to higher altitudes which dilutes the aerosol loading in the boundary layer in summer (in other words, in winter aerosols are confined into a thinner mixing layer than in summer, which leads to a higher particle concentration in winter (Ding et al., 2016a); 3) both in-cloud scavenging and precipitation scavenging are stronger in summer than winter because of more precipitation in summer than in winter.

Contrary to the otherwise relatively low scattering coefficients in summer, the average $\sigma_{sp}$ in June (534 Mm$^{-1}$) was the third highest among all 12 months, just below 618 Mm$^{-1}$ in December and 603 Mm$^{-1}$ in January. The median $\sigma_{sp}$ in June was 469 Mm$^{-1}$, even higher than the median value of December (463 Mm$^{-1}$). Moreover, in June the 10[th] and 25[th] percentiles of $\sigma_{sp}$ were the highest among all 12 months. This indicates that the high monthly average scattering coefficient was not caused by some short episodes. Absorption coefficient was only a little bit higher than in the adjacent months (Figure 3b). However, the average SSA in June was 0.95 at 520 nm, much higher than in any other month. Both backscatter fraction (b) and SAE reached the lowest monthly averages in June, even though their seasonal variation was otherwise different. The independent parameters $PM_{2.5}$ and the geometric mean diameter and the volume mean diameter were also higher in June than in the adjacent months, both of which are consistent with the higher $\sigma_{sp}$ and the high mean diameters also with the low *b* and *SAE*. The high $\sigma_{sp}$ and $PM_{2.5}$ values in June were possibly due to biomass burning since in June there were more fire spots observed than in any other month within a 300 km range from SORPES (Ding et al., 2013a, 2013b). A more detailed discussion of the period is omitted from the present paper, however.

### 3.3 Diurnal cycles

The hourly-averaged data were classified according to the hour of the day in the four seasons. Figure 4 presents diurnal cycles of aerosol optical properties $\sigma_{sp}$, $\sigma_{ap}$, SSA, and *b* as well as the diurnal cycles of the supporting parameters photochemical age and the particle number size distributions. The averaging was conducted so that the value for 00:00 is the average of 5-min data between 00:00 and 01:00 and analogically for all hours of the day. All times discussed in this section are local time.


A clear diurnal variation of $\sigma_{sp}$ and $\sigma_{ap}$ was observed in all seasons. In each season, there was an evident minimum in the afternoon around 14:00 and a continuous almost flat peak from 20:00 to 8:00 the next day. Within these 12-hour flat peaks, there were two maxima, one around 20:00-23:00 in the evening and another at around 8:00. We call them the 'evening peak' and the 'morning peak' here. Horvath et al. (1997) and Lyamani et al. (2010) found a similar cycle with two maxima

in wee hours and afternoon. Notably, even though the minimum in the early morning is only 0~5% percent lower than the two peaks, we use two maxima and two minima to divide the whole cycle into four stages: 1) in the early morning, SSA started to decrease and reached a minimum around 7:00 and 8:00. This decrease indicates that relatively more light absorbing aerosol was emitted around that period and possibly influenced by vehicle emission during rush hours. 2) Then SSA increased significantly and reached the highest value of the day at around 14:00. The SSA at $\lambda$ = 520 nm even

exceeded 0.94 in summer and spring. Secondary aerosols formed by gas-to-particle conversion processes, for instance sulfates and nitrates were a likely cause for such a high SSA. 3) After reaching the maximum, SSA decreased rapidly and reached the lowest daily values at around 19:00-20:00, which can be considered as a combined influence by deceleration of secondary aerosol formation at dusk and the addition soot emission during rush hours. 4) SSA increased gradually from 20:00 to early morning the next day.

The diurnal cycles are influenced by the variations of the PBL height, anthropogenic activities and photochemical reactions. The strong decrease in $\sigma_{sp}$ and $\sigma_{ap}$ after the morning peak can be associated with the boundary layer development in daytime, which enhances convective activity and decreases the particle loading at ground level. Both $\sigma_{sp}$ and $\sigma_{ap}$ reached minima at around 14:00 and then increased again as vertical mixing got weaker. The increase of absorption in the

afternoon was approximately as fast as the decrease after the morning peak: $\sigma_{ap}$ at 20:00 was almost the same as at 8:00. On the other hand, the increase of scattering after 14:00 was somewhat slower than the decrease from 8:00 to 14:00. The maximum $\sigma_{sp}$ was reached later than the maximum $\sigma_{ap}$ and as a result SSA decreased to a minimum at about 19 :00– 20:00.

The diurnal cycle of SSA followed very closely the diurnal cycle of the photochemical age: the air masses with the lowest

photochemical age contained the aerosol with the lowest SSA which suggests that in these air masses there were BC particles that had not been coated with as thick a coating with light-scattering material as in the aged air masses. The most probable source for such intensive $NO_x$ emission in the morning is the rush-hour traffic. For stage 2, we notice that the start time for the increase is one hour earlier in summer and spring than in winter and autumn. An opposite time difference can be observed in the evening. The obvious explanation is the seasonal variation of the length of the day: the average

sunrise (and sunset) times are approximately 7:00 (17:30), 5:30 (18:30), 5:15 (19:00) and 6:15 (19:30) for winter, spring, summer and autumn, respectively.

The diurnal cycle of the size distribution offers additional data to explain the SSA cycle. On the average, at about 10:00-12:00 (Figure 4f) when the photochemical reactions are active and solar radiation is strong, new particle formation (NPF)

occurred at SORPES as discussed in detail by Qi et al. (2015). NPF produces small particles that are initially too small to affect the total light scattering. However, at the time of the NPF some of the older, larger particles still remained which



resulted in a bimodal size distribution with a fast growing nucleation mode and an Aitken mode in the particle diameter range of about 70–90 nm. The species that condense on the newly-formed particles are typically light scattering inorganic and organic species and they also condense on the Aitken-mode particles. At about 12:00 – 14:00 the newly-formed particles had grown into size range of about 20 – 50 nm, which has still a very small scattering cross section ($C_s$ =

$Q_s(\pi/4)D_p{}^2$, where $Q_s$ is the scattering efficiency) compared with that of the larger mode that at this time had grown by condensation of light-scattering species to ~100 nm. This mode was now responsible for the high SSA. These particles were also associated with the photochemically aged air masses. In the afternoon, the boundary layer started to decrease and the air masses contained less aged and more fresh aerosols and lower SSA. The minimum photochemical ages and SSA were observed at stage 3 at about 18:00 – 20:00 LT, depending on the season. At that time the size distributions show

that the number concentrations of Aitken-mode particles increased more than by a straightforward growth of the particles formed earlier during the day. This suggests that there was an injection of fresh BC particles into the boundary layer during the evening rush hour. Very probably aerosol was then an external mixture of those grown after the NPF and the more freshly-emitted BC. At stage 4, during the course of the night $\sigma_{ap}$ decreased slowly indicating weaker BC emissions. However, the particles kept growing both by condensation and coagulation as seen by the growing GMDs and $\sigma_{sp}$. Also

SSA grew all night long which suggests that the main mechanism of the growth was again condensation of light-scattering species. The formation of these species in the absence of light may be due to $NO_3$ radical chemistry, as was suggested to be the explanation of a similar increase of SSA at night in Sao Paolo, Brazil (Backman et al., 2012).

The diurnal cycle of SAE, a usual qualitative indicator for the dominating particle size, suggests that the size was the

smallest during 14:00 – 16:00 when SAE reached maximum values of each seasonal diurnal cycle, approximately 1-2 hours later than SSA. The interpretation of the inverse relationship between SAE and particle size is not correct when the size distribution consists of multiple modes (e.g., Schuster et al., 2006). This is consistent with the comparison between the diurnal cycles of the GMD and SAE: the GMD was the smallest during 10:00 – 12:00 when the newly-formed particles formed the nucleation mode at the same time the Aitken mode was present. After about 16:00–18:00 the size distribution

became close to unimodal and the relationship inverse: GMD grew almost steadily and SAE decreased almost steadily during the night until the morning peak. Then the $\sigma_{ap}$ grew and GMD decreased but so did SAE still until about 8:00 of the next day.

The relationship of the backscatter fraction $b$ and particle size is basically similar to that of SAE and particle size: for

small particles $b$ is large - for Rayleigh scattering, i.e., for gas molecules $b = 0.5$ - and it decreases with increasing particle size. For real atmospheric size distributions, the interpretation of $b$ gets complicated when multiple modes are present, as for SAE. The diurnal cycles of $b$ suggest that particles were the largest just before the morning peak at about 5:00 after which the optically dominating particle size decreased and $b$ grew until 12:00 – 19:00 depending on season. After the peak value $b$ decreased almost steadily during the night in agreement with the growing GMD, in principle very much like

SAE. There are also some obvious differences between $b$ and SAE diurnal cycles. The $b$ peak values were reached later than SAE in winter, spring, and autumn but earlier in summer. Another difference is that in summer $b$ remained at a steady



high value for several hours, from 12:00 to 21:00 contrary to SAE.

The range of AAE values during the diurnal cycle was not very large. AAE was usually the smallest during the morning peak and then it grew slowly. The largest diurnal growth was in spring: from ~1.00 to ~1.09. The average AAE diurnal
variation is in a range that suggests it can be attributed to variations of the diameter of a BC core and a light-scattering shell coating it (e.g., Gyawali et al., 2009; Lack and Cappa, 2010; Wang et al.,2016). A detailed analysis of this would require measurements with an instrument that can actually measure the BC core and shell such as the single-particle soot photometer (SP2) (e.g., Schwarz et al., 2008).

**3.4 Aerosols in different air masses**
Below we will study how the aerosol optical properties vary simply as a wind blows from various directions. More detailed analyses are conducted by backward Lagrangian dispersion modeling and backtrajectory cluster analysis.

**3.4.1Aerosol optical properties as a function of wind speed and direction**
Figure 5a and b show the scattering and absorption coefficients and Figure 5c gives the photochemical age as $-\log(NO_x/NO_y)$ observed at the different wind directions and speeds by using polar coordinates. Figure 5d presents a standard wind rose for the 2-year measurement period. The prevailing wind direction is E and NE. Only a small fraction of wind blew from SW and NW. There was no big difference in wind speed from the different directions: the largest and lowest average WS was 3.9 m/s and 2.0 m/s from the directions 100~105°and 160~165°, respectively.

Generally, both $\sigma_{sp}$ and $\sigma_{ap}$ decreased with increasing wind speed when WS> 1 m/s in all directions (Fig 5a and 5b). This suggests that at strong winds air was generally more diluted with cleaner air from upper altitudes by turbulent mixing. An exception was the west-northwesterly (WNW, WD ≈ 285 ± 15°) sector since $\sigma_{sp}$ did not decrease much with increasing wind. In this sector the above-mentioned dilution effect required stronger winds: the highest $\sigma_{sp}$ was observed at WS ≈
2.5 ± 0.5 m/s but almost as high values were still observed at WS ~7 m/s above which $\sigma_{sp}$ started decreasing. However, the frequency of winds from this sector WD was very low, only 1.3% of whole period so it does not change the general picture of decreasing scattering with increasing wind. High $\sigma_{sp}$ (> 560 Mm$^{-1}$) was also observed at weak (WS < 2 m/s) southwesterly (SW, WD ≈ 215 ± 10°) winds. This is probably a mixture of large-scale, regional and local emissions. The center of the city is in that direction and at low wind speeds pollutants are easily transported to SORPES without a strong
dilution. However, the center of Nanjing is still too close to be the most important contributor to light-scattering particles. A rough estimate of the transport distance needed to grow particles to sizes that scatter light efficiently can be calculated by using the information on the particle growth rate and wind speed. The time evolution of the average diurnal particle size distribution (Figure 4f) shows that new particles with $D_p ≈$ 10 nm formed before noon grew to sizes that scatter light significantly, i.e. $D_p >$~100 nm in about 10 ± 1 hours yielding an approximate growth rate of GR ≈ 9 ± 1 nm h$^{-1}$ which
agrees with the analysis of Qi et al. (2015). In 10 hours at WS = 0.5 m/s the air mass would have drifted a distance of 18 km which is just the distance to the city center. At the weakest winds from the SW the contribution of the city to the





amount of scattering particles may thus have been significant. When wind speed is higher, regional transport of pollutants plays probably a more important role. The highest $\sigma_{sp}$ in Figure 5a is at WD $\approx 285 \pm 15°$ and WS$\approx 2.5 \pm 0.5$ m/s. At this speed the air masses drift $90 \pm 18$ km in 10 hours that was estimated to be the time to reach optically significant particle sizes since the formation, suggesting that the aerosol responsible for these highest values originated from outside the

urban area of Nanjing. However, the above-described transport distance estimate applies only to cases when aerosol was formed by NPF and grown then by condensation and coagulation. Primary particles such as BC are emitted from vehicles in the size range of $\sim 60 \pm 20$ nm (e.g., Bond et al., 2013). At GR $\approx 9$ nm h$^{-1}$ it would take $\sim 5 \pm 2$ hours to grow them to sizes $> 100$ nm. At low and moderate winds (WS $< \sim 3$ m/s) from east and southeast (WD range $\sim 75° - 165°$), $\sigma_{sp}$ was also high, $> 480$ Mm$^{-1}$, suggesting that the air masses from YRD have a higher $\sigma_{sp}$ than the average value.

The absorption coefficient had a different dependence on wind. The relationship between $\sigma_{ap}$ and WS is very clear: at weak wind $\sigma_{ap}$ was obviously high and it decreased significantly as WS increased in all wind directions. It was the highest at weak southwesterly winds (SW, WD $\approx 225 \pm 15°$ and WS $\approx 0.8 \pm 0.2$ m/s) but high $\sigma_{ap}$ values were especially observed at winds from the southern sector WD $= 180 \pm 30°$ at WS $<2.5$ m/s. The high $\sigma_{ap}$ values in the SSW sector (WD $\approx 200 \pm$

$10°$) were observed approximately from the same direction as the above-mentioned $\sigma_{sp}$ SW peak direction and can be attributed to emissions from the urban areas of Nanjing. This is also supported by the comparison with the polar contour plot of the photochemical age of air masses, i.e., $-\log(NO_x/NO_y)$ (Figure 5c). It shows that photochemically fresh air with $-\log(NO_x/NO_y) \le 0.12$ was observed at SORPES with winds from this same direction. Also $\sigma_{ap} \ge 42$ Mm$^{-1}$ from the SSE (WD $\approx 165 \pm 15°$) follows approximately the contour of $-\log(NO_x/NO_y) \le 0.12$ suggesting that the high $\sigma_{ap}$ from this

sector was associated with photochemically fresh traffic emissions. The wind direction does not point to the center of the city so it is not that obvious where it comes from.

Photochemically fresh air with $-\log(NO_x/NO_y) \le 0.2$ was observed also with winds from the same WNW sector as the highly scattering aerosol. These are controversial results since to grow particles to the size range that scatters light at high

efficiency requires time. Further, in the air masses from this sector BC concentrations and thus $\sigma_{ap}$ were low which suggests the NO$_x$ observed did not come from sources that emit also high BC concentrations. Another interesting wind direction is the NE sector (WD $\approx 45 \pm 15°$). At all wind speeds from that sector the photochemical age is relatively low and the lowest ($-\log(NO_x/NO_y) \le 0.12$) at high wind speeds (WS $\approx 6 \pm 1$ m/s). With this WD, WS combination both $\sigma_{sp}$ and $\sigma_{ap}$ were low. This suggests that in this direction there is a NO$_x$ emitter that does not emit significant amounts of BC

and that it is relatively close since particles have not grown to size ranges large enough to affect $\sigma_{sp}$ significantly.

### 3.4.2 Lagrangian dispersion modeling

The 72-hour retroplumes were calculated every 3 hours, so there were 8 retroplumes/day. To assess the source areas of high (low) $\sigma_{sp}$ and $\sigma_{ap}$ the retroplumes of the three days with the highest (lowest) daily-averaged $\sigma_{sp}$ and $\sigma_{ap}$ of each month

were averaged, alogether 576 retroplumes. Since there are approximately 30 days/month the three lowest daily averages represent approximately the lowest 10% and the three highest daily averages the highest 10% of the daily averages. The




reasoning for this approach is that 1) diurnal cycles are dominated by boundary layer height cycles and photochemistry-driven cycles, and 2) seasonal variation hides obvious pollution episodes in cleaner months. The results are shown in Figure 6. It shows that the potential source region of highest 10% of both $\sigma_{sp}$ and $\sigma_{ap}$ are within a large area from the eastern China, with the highest retroplume around Nanjing spread between the longitudes 115°E ~ 123°E (~700 km) and

the latitudes 28°N ~ 35°N (~800 km) (Figure 6a and 6b). For the lowest 10% of both $\sigma_{sp}$ and $\sigma_{ap}$ daily averages air masses mainly originated from the ocean in the east with a fast transport pathway (Figure 6c and 6d). A comparison of Figs. 6a and 6b suggests a slightly different transport pattern for the highest 10% of $\sigma_{sp}$ and $\sigma_{ap}$ daily averages. For both $\sigma_{sp}$ and $\sigma_{ap}$ sub-regional air masses from the southeast contribute clearly to highest 10%, i.e. from the city cluster in Yangtze River Delta region (Ding et al., 2013b) as well as air masses transported from various direction, indicating more local

emissions for Nanjing and the adjacent cities. The most obvious difference is that high $\sigma_{sp}$ is more clearly associated with air mass transport from the northwest, especially from regions north of 35°N, i.e. from the North China Plain and Shandong province,. These are regions where the high emission of $SO_2$ could have a large regional impact on scattering coefficient in the south as high concentration of sulfate is formed.

The results from retroplume calculation (Fig. 6) seems apparently inconsistent with the results from wind rose analysis (Fig. 5). In fact, the wind rose analysis was based on hourly data but the retroplume calculations are based on daily averages. In other words, the wind rose results show more details on the change of aerosols coefficients according to change of wind direction of local wind at high temporal resolution, but the latter shows more information about the history of air masses when they get transported long distances.

### 3.5 Analysis of a polluted winter period

The above-presented analyses were made by using the whole data set. Below a winter-time polluted period from 1 November 2013 to1 March 2014 will be discussed in detail. Tang et al. (2016) analyzed the sources contributing to submicron particulate matter in the haze episodes observed in the center of Nanjing in December 2013 and concluded

that the high aerosol pollution was mainly due to regional transport. The analysis below complements that by Tang et al. (2016) since they did not present an analysis of the evolution of AOPs.

The backtrajectories were clustered into four clusters as explained in section 2.5. The average trajectories of the clusters obtained for the period are presented in Figure 7. The clusters are given more descriptive names: cluster 1 is YRD that

constitutes 36 % of the trajectories and represents air from the east; cluster 2 is COASTAL, 31 % of the trajectories, representing air coming from the northeast partly over the sea at a low speed; cluster 3 is WEST, 23 % of the trajectories, representing purely continental air from the west; cluster 4 is NORTH, 11 % of the trajectories, representing fast-flowing air from the north. In addition to the average trajectories of each cluster, also the average 72-hour retroplume is depicted in Figure 7 as the background color. It is reasonable as it resembles the retroplumes for the highest 10% daily averages

of $\sigma_{sp}$ and $\sigma_{ap}$ (Fig. 6a and 6b) for the whole period.





The time series of several extensive and intensive aerosol properties within the winter period are presented together with the modeled planetary boundary layer height (PBLH) in Figure 8. In addition, the time series of the trajectory cluster classes are presented on the top line with colors.

### 3.5.1 Extensive aerosol properties

It is possible to count approximately 14 – 16 distinguishable pollution episodes by using either the PM concentration or the scattering coefficient time series. The analyzed period was 120 days long so on the average there was a pollution episode about every $8 \pm 1$ days. The definition of the start and end of a pollution episode is not unambiguous, however, so the above number should be treated cautiously. The concentrations of $PM_{2.5}$ and $PM_1$ tracked each other very closely suggesting that within the uncertainties most of the aerosol mass is secondary, pollution-related compounds, and that soil dust which in general is in super-micron size range contributed very little to aerosol mass and scattering at SORPES.

The trajectory cluster time series, color coded on the top of Figure 8, shows that when air masses were associated with the WEST or the YRD clusters there were no big differences in concentrations. The COASTAL trajectory cluster was often but not always associated with lower concentrations. Most of the episodes ended with trajectories associated with the NORTH cluster. Meteorological analyses show that the trajectories associated with the NORTH cluster brought air from the north, high above Beijing during cold fronts. There were also episodes during which there were trajectories belonging to many different clusters, YRD, WEST and COASTAL but there were no clear differences in concentrations until the clearing phase associated with the cluster NORTH. During these episodes polluted air arrived from all directions in line with wind roses that showed there was no strong dependence on wind direction. Instead, the concentrations kept rising.

Most episodes followed a similar pattern: during the evolution phase the PM concentrations grew day after day during several days at a rate of some tens of µg m$^{-3}$ / day but the end of the episode was usually abrupt, air cleared within hours. The largest drop in the period occurred on 2 – 3 February when PM concentrations, $\sigma_{sp}$ and $\sigma_{ap}$ decreased by more than an order of magnitude within hours. The same cycle applied to all extensive parameters: PM and $BC_e$ concentrations, $\sigma_{sp}$ and $\sigma_{ap}$ increased clearly more slowly during the growth phase of the episodes than decreased in the end (Figure 8b and 8e). At the same time the daily maximum PBLH (Figure 8a) decreased during most of the episodes depicted in Figure 8 from more than ~1500 m to less than ~700 m. This PBLH decrease is in agreement with the analysis of Petäjä et al. (2016) and Ding et al. (2016a) who showed that high PM and especially BC concentrations enhance the stability of a polluted boundary layer, which in turn decreases the boundary layer height and consequently cause a further increase in PM concentrations.

Even during the growing phase of the episodes there were obvious diurnal cycles of the AOPs. For instance, low PM concentrations, $\sigma_{sp}$, and $\sigma_{ap}$ during daytime and higher at night combined with a growing trend can be explained with the formation of a residual layer: when the PBLH decreases at night part of the aerosol remains above the PBL. The following





day new pollutants get mixed with the pollutants remaining in the residual layer. This leads to a continuous accumulation of aerosols in the PBL and a slower, non-symmetric cycle. Part of the accumulation is due to BC, as is seen in the increasing $BC_e$ concentration even though the mass fraction of $BC_e$ ($f_m(BC_e) = BC_e/PM_{2.5}$) decreased from ~10% to ~2 % during the growth phase of the episodes (Figure 8c). The particle number size distribution time series (Figure 8d) shows

that there are indications of new particle formation (NPF) also during the polluted period and that the new particles grew quickly into optically relevant size ranges and thus contributed to visibility reduction. Kulmala et al. (2016) estimated that about half of the particles in the accumulation mode, i.e., the optically relevant size range originate from NPF at SORPES even though the pollution level is high.

**3.5.2 Evolution of intensive aerosol properties**

Also the intensive aerosol properties, i.e., those that do not depend on the amount of particles, clearly evolved during the pollution episode cycle. First, the effective particle size grew which is depicted as the time series of the geometric mean diameter GMD (Figure 8d). There was an obvious diurnal cycle with the GMD as well. The growth leads to many changes in the intensive AOPs. The mass scattering efficiency MSE = $\sigma_{sp}/m$ (Figure 8f) grew during the extended pollution

episodes from about 4 $m^2$ $g^{-1}$ to ~6 $m^2$ $g^{-1}$ which is in the range presented by Hand and Malm (2007). In other words, the unit mass of aerosol scattered light more efficiently at the end of the episode than at the beginning. An obvious explanation is that this is due to the growth of both particle diameter and the scattering efficiency ($Q_s$) even though also the changing refractive index plays a role in this. The Ångström exponents of scattering and absorption (SAE and AAE) (Figure 8g) as well as the backscatter fraction $b$ (Figure 8i) decreased as the particles grew. The decrease of AAE during the growing

phase of the episodes are explainable by a growing shell on a BC core as has been modeled by Gyawali et al. (2009) and Lack and Cappa (2010). At the same time SSA (Figure 8h) increased which can be explained by condensation of light-scattering material and thus increasing the thickness of a shell surrounding a BC core.

Contrary to the strong changes of the other AOPs during the growth phase of the pollution episodes, the radiative forcing

efficiency (RFE = $\Delta F/\tau$) (Figure 8j) did not vary strongly. This is interesting since intuitively it could be thought that the higher SSA grows the lower is the RFE, in other words the more do the particles cool the atmosphere. That RFE remained fairly stable is due to the growth of the particles: larger particles scatter light upwards less efficiently than small ones which to some extent compensates the higher SSA. A similar phenomenon was observed by Garland et al. (2008) during an intensive campaign in Guangzhou in southeast China in July 2006.

**3.6 Relationships of AOPs**

Above the evolution of AOPs during the polluted winter period were discussed, here the whole 2-year data set will be used for an analysis of the relationships of AOPs. First some of them are compared with the effective mean diameters obtained from the DMPS data, next some dependencies of AOPs on each other and the photochemical age are discussed

and in the end there is an analysis of the radiative forcing efficiency.





### 3.6.1 Relationships between optical properties and particle size

The scatter plots (Figure 9) show that all the analyzed AOPs clearly depend on SMD and VMD but not so obviously on GMD. $\sigma_{sp}$ was generally the higher the larger the weighted diameters were (Figure 9a). This is the intuitively most logical relationship of all those presented in Figure 9 since $\sigma_{sp}$ of a size distribution is calculated from (12) that includes the

surface area of a spherical particle. When particles grow their surface area grows and they scatter more light.

The observed darkest aerosol had SSA< 0.85 which is not even close to that of pure fresh BC. Then GMD was in the range of $30 - 80$ nm, SMD at 250 nm, VMD at $300 - 350$ nm. These can be compared with BC size distributions observed elsewhere. Schwarz et al. (2008) measured BC size distributions with a single-particle soot photometer (SP2) and found

that the mass median diameter (MMD) and geometric standard deviation of these distributions were 170 nm and 1.71, respectively, in an urban air, and 210 nm and 1.55, respectively, in a continental background air. These values yield number mean BC diameters of 72 nm and 118 nm for urban and continental background air. Our GMD was in the same range but VMD clearly larger. It can therefore be deduced that even the darkest aerosol we observed was not fresh BC. The cases with very high SSA and GMD < 40 were very probably associated with NPF events. The SSA growing with

growing weighted mean diameters is plausibly explainable by a larger scattering shell on an absorbing core.

SAE is a parameter that is often used as qualitative indicator of dominating particle size so that large values indicate a large contribution of small particles and small values a large contribution of large particles. For SMD and VMD this is indeed so in our data (Figure 9c). GMD, on the other hand, cannot be predicted at any uncertainty with SAE. All these

are consistent with the relationships observed in a completely different environment, the boreal forest at SMEAR II, Finland (Virkkula et al., 2011). All the same conclusions apply to the relationship between the weighted mean diameters and backscatter fraction, *b* which is an even better indicator of the dominating particle size (Figure 9d). This is logical since it is well known that forward scattering increases with growing particle size. Both *b* and *SAE* provide information on the particle size distribution but they are sensitive to somewhat different particle size ranges (e.g., Andrews et al., 2011;

Collaud Coen et al., 2007). Collaud Coen et al. (2007) presented a detailed model analysis of both of these AOPs and showed that *b* is most sensitive to small accumulation mode particles, i.e., particles in the size range < 400 nm whereas *SAE* is more sensitive to particles in the size range $500 - 800$ nm. Delene and Ogren (2002) showed the importance of fine-to-coarse scattering ratio (= ratio of scattering in the $PM_1$ and $PM_{10}$ size ranges).

The relationships of SSA are analyzed further with the scatter plots in Figure 10. SSA was in general the higher $\sigma_{sp}$ was but the range of SSA was very large, ~$0.82 - 0.98$ when $\sigma_{sp}$ was in the range < 400 Mm$^{-1}$(Figure 10a). SSA was always high, > 0.94 in the most polluted air masses when $\sigma_{sp}$ was > 1000 Mm$^{-1}$. The color-coding with the photochemical age further shows that the most polluted air masses were mainly aged with PA > 0.4. There are, however, some data points with PA < 0.2 indicating relatively fresh air even at $\sigma_{sp}\approx$1000 Mm$^{-1}$. These points have a lower SSA, down to ~ 0.90. It is

very unlikely that scattering coefficients as high as 1000 Mm$^{-1}$ are due to nearby emissions so the data suggests that in these cases the aerosol consists of an external mixture of long-range transported strongly scattering particles and fresh,



possibly traffic-related BC particles that have not yet been coated with a thick shell. The comparison of SSA with the backscatter fraction $b$ (Figure 10b), shows that the lowest SSA was mainly observed when $b$ was in the range ~0.12 – 0.16 and the highest SSA at $b < 0.10$ which indicates large particles as discussed above. The highest backscatter fractions were observed in fresh air masses (PA < 0.2) and lowest in the most aged air masses. However, there were again some

points with both a low PA and $b$ suggesting that there was an external mixture of long-range-transported and fresh aerosols. The scatter plot of SSA vs. PA (Figure 10c) shows very clearly that the darkest aerosol (SSA < 0.9) was in the freshest air masses. But it also shows that in some of the fresh air masses with PA < 0.1 SSA was very high, > 0.96 suggesting that there were also such $NO_x$ emissions that were not associated with BC emissions. This is in line with the analysis presented above, in section 3.4.1.

The comparison of SSA with (Figure 10d) shows that AAE was in the range ~0.9 – 1.2 which is approximately that estimated from traffic emissions (e.g., Zotter et al., 2017). Lower AAE was observed with high SSA and low backscatter fractions, i.e., when particle size distributions were dominated by large particles. In the analysis of the polluted winter period it was stated that the decrease of AAE during the growing phase of the episodes could be explained by a growing

shell on a BC core (e.g., Gyawali et al., 2009; Lack and Cappa, 2010). Similar relationship is observed in the scatter plot of the whole data set.

### 3.6.2 Analyses of the radiative forcing efficiency

The parameters aerosol-related affecting the radiative forcing efficiency RFE = $\Delta F/\tau$, Eq. (9), are the single-scattering

albedo and backscatter fraction. A simple intuitive assumption is that the darker the aerosol is the less it cools the atmosphere, which means the RFE should be higher. However, the scatter plot of the real data shows there was a very weak decreasing relationship between RFE and SSA alone (Figure 11a). The relationship between RFE and $b$ is somewhat clearer: RFE decreases with increasing $b$ (Figure 11b). But the data are really scattered in a wide range which is due to that RFE depends on both SSA and $b$.

The relationships between of RFE, SSA and b become clearer in a scatter plot of SSA vs. b where the isolines of constant RFE as a function of both parameters are plotted (Figure 12). Now it is found that a lot of the data points are between the RFE isolines -22 W m$^{-2}$ and -28 W m$^{-2}$. The data don't exactly get clustered around any single RFE isoline. This was studied by classifying SSA into backscatter fraction bins with a width of $\Delta b = 0.01$, calculating the percentiles of the

cumulative distributions in each bin and presenting them in the box plots in Figure 12. Interestingly, the highest $b$ bin ($b$ = 0.17-0.18) has the lowest median SSA but also the lowest RFE, ~ -28 W m$^{-2}$, but the lowest b bin ($b$ = 0.09-0.10) has the highest SSA but the highest RFE, ~ -24 W m$^{-2}$. This is systematic: decreasing $b$, i.e., growing particles resulted in a higher SSA and RFE. This means that actually the darker aerosol had a higher negative radiative forcing efficiency, i.e., they cool the atmosphere more efficiently than the aerosol with the higher SSA. This is due to the size of the particles:

small particles scatter light upwards more efficiently than the big ones which to some extent compensates the darkness of them.



The probability distribution of RFE is presented in Figure 11 and the percentiles of the cumulative distribution of RFE in Table 1. The median RFE = -24.7 W m$^{-2}$ and the $10^{th}$ to $90^{th}$ percentile range from -27.1 to -22.1 W m$^{-2}$. Again the values are compared with those calculated for the boreal forest site in Finland (Virkkula et al., 2011). At SMEAR II the median RFE = -23.4 W m$^{-2}$, higher than at SORPES and the $10^{th}$ to $90^{th}$ percentile range is from -28.8 to -16.9 W m$^{-2}$, which is a

clearly larger range of values. The main differences between these two sites are that at the polluted site RFE is lower, i.e., the aerosol cools the atmosphere more efficiently and that the distribution of RFEs are more narrow than at the clean site. This may be due to the above described relatively constant RFE during the evolution of the pollution episodes which occurred frequently. This is in line also with the climatology shown by Andrews et al. (2011) according to which the cleanest sites had the widest RFE range.

### 4. Summary and Conclusions

Aerosol optical properties (AOPs) were measured at SORPES, a regional background station in Nanjing. In this study we have presented basic descriptive statistics, seasonal and diurnal variations, studies of transport and relationships between the AOPs and some supporting data during the two-year period from June 2013 to May 2015.

On the average aerosol was highly scattering with an average $\sigma_{sp}$ = 410 ± 320 Mm$^{-1}$ and $\sigma_{ap}$ = 26 ± 19 Mm$^{-1}$ and single-scattering albedo 0.93 ± 0.03 for the green light. A comparison showed that the SSA of aerosol is slightly higher in Nanjing than that published from most other sites in China and elsewhere. In the comparison we included also other published data from other inland sites within the YRD region and also there SSA appears to be higher than at most other sites. This

suggests that the concentrations of the condensable vapors that make the aerosol grow may be higher within the source areas of aerosol influencing Nanjing than at the compared sites. However, the presented comparison has several sources of uncertainty: the data from the different sites did not cover equally long periods, the sampling protocols and instruments at the different sites were not similar and the data were processed with different algorithms. To get a reliable comparison of the aerosol at the different locations in China all methods should be harmonized and quality controlled.

The extensive AOPs had typical seasonal cycles with high scattering and absorption coefficients in winter and lower in summer: the averages were $\sigma_{sp}$ = 545 ± 425 Mm$^{-1}$ and $\sigma_{ap}$ = 36 ± 24 Mm$^{-1}$ in winter and $\sigma_{sp}$ = 364 ± 294 Mm$^{-1}$ and $\sigma_{ap}$ = 20 ± 13 Mm$^{-1}$ in summer. The intensive AOPs had no clear seasonal cycles; the variations of them were rather related to the evolution of pollution episodes. The diurnal cycles of the intensive AOPs were clear. The diurnal cycles of SAE and

$b$ suggest that particles were the largest just before the morning peak after which the optically dominating particle size decreased and $b$ grew until afternoon or evening, depending on season. After the peak value $b$ decreased almost steadily during the night in agreement with the growing GMD, in principle very much like SAE. So the diurnal cycles of SAE and $b$ were consistent with the cycle of the particle number size distribution. The differences in the amplitude of the variations and in the timing during the four seasons were very probably due to the variations in the solar radiation and the consequent

gas–to–particle phase transition.



SAE is generally used as qualitative indicator of dominating particle size with an inverse relationship between SAE and size. For surface mean diameter SMD and volume mean diameter VMD this was indeed so. The geometric mean diameter GMD, on the other hand, did not correlate at all with SAE. An explanation for this is that the particle number size distributions are dominated by so small particle sizes that their contribution to light scattering is negligible. All these are

consistent with the relationships observed in a completely different environment, the boreal forest at SMEAR II, Finland. All the same conclusions apply to the relationship between the weighted mean diameters and backscatter fraction, $b$ which proved to be a slightly better indicator of the dominating particle size, there was even some – although weak – negative correlation between $b$ and GMD.

The average AAE for the wavelength range 370 – 950 nm was 1.04 and the AAE range 0.7 – 1.4. These AAE values can be explained with different amounts of non-absorbing coating on pure BC cores and different core sizes so the data does not suggest any significant contribution to absorption by brown carbon which would result in a higher AAE. Note, however, that these values were calculated with a non-wavelength-dependent multiple-scattering correction factor $C_{ref}$. Algorithms that assume wavelength-dependent $C_{ref}$ would yield higher AAE and would also lead to a conclusion of larger

contribution by brown carbon. Comparison of the published AAEs is difficult since some authors have used wavelength-dependent $C_{ref}$ and some not. Since no unambiguous proof of this in either direction has been given, the uncertainty of AAE is high.

The source areas were studied by comparing the AOPs with the local wind, by backward Lagrangian dispersion modeling

and by a backtrajectory cluster analysis. High $\sigma_{sp}$ was observed at all wind directions. By using the observed particle growth rates and local wind it could be estimated that the center of Nanjing is too close to be the most important contributor to light-scattering particles. On the other hand, the comparison of $\sigma_{ap}$ with the local wind and with the photochemical age of air masses suggests that high concentrations of light-absorbing aerosol, mainly BC, originated from the urban areas of Nanjing and near-by traffic emissions. For the Lagrangian dispersion modeling, the retroplume analysis,

the daily-averaged $\sigma_{sp}$ and $\sigma_{ap}$ were used to eliminate the effect of diurnal variations, mainly due to the varying boundary layer height. The result of the retroplume analysis is that the sources of high $\sigma_{sp}$ and $\sigma_{ap}$ are within a large area, it is not possible to pinpoint single sources for the high values with this method. The area that is the main contributor to the highest quartile is large. The distance from west to east between the longitudes 115°E and 123°E at latitude 30°N is about 700 km and from south to north between latitudes 29°N and 35°N about 800 km.

In pollution episodes the daily maximum PBLH decreased in agreement with the analysis of Petäjä et al. (2016) and Ding et al. (2016a) who showed that high PM and especially BC concentrations enhance the stability of a polluted boundary layer, which in turn decreases the boundary layer height and consequently cause a further increase in PM concentrations. In these episodes, PM concentrations, $\sigma_{sp}$ and $\sigma_{ap}$ typically increased gradually and remained high during several days but

decreased faster, sometimes even by more than an order of magnitude within some hours and remained at the lower level for a shorter period than at the higher level. Most of the episodes ended with trajectories associated with the trajectory





cluster that brought air from the north, high above Beijing during cold fronts. During the growth phase of the pollution episodes also the intensive aerosol optical properties evolved clearly. The mass scattering efficiency MSE of of PM$_{2.5}$ grew during the extended pollution episodes from ~4 m$^2$ g$^{-1}$ to ~6 m$^2$ g$^{-1}$. In other words, the unit mass of aerosol scattered light more efficiently at the end of the episode than at the beginning. The mass fraction of BC$_e$ decreased from ~10% to

~2 % during the growth phase of the episodes. The growth of the particles also clearly lead to the decrease of the Ångström exponents of scattering and absorption (SAE and AAE) and the backscatter fraction $b$ and to the growth of SSA. This further lead to a higher radiative forcing efficiency RFE. In other words, the darker aerosol – the aerosol that had a higher BC mass fraction – had a more negative RFE, i.e., they have the property of cooling the atmosphere more efficiently per unit optical depth than the aerosol with the higher SSA and a lower BC mass fraction. This counterintuitive result is due

to the size of the particles: the upscatter fraction of small, more fresh BC particles is higher than that of larger aged ones which more than compensated the darkness of them. The RFE probability distribution at SORPES was clearly more narrow than at a clean background site which is in agreement with a published RFE climatology.

**Acknowledgements**

The research was supported by the Jiangsu Provincial Natural Science Fund (No.BK20140021), National Science Foundation of China (D0512/91544231, D0512/41422504) and National Key Research and Development Program of China (2016YFC0200500), and Academy of Finland's Centre of Excellence program (Centre of Excellence in Atmospheric Science – From Molecular and Biological processes to The Global Climate, project no. 272041) .

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



**Figures**

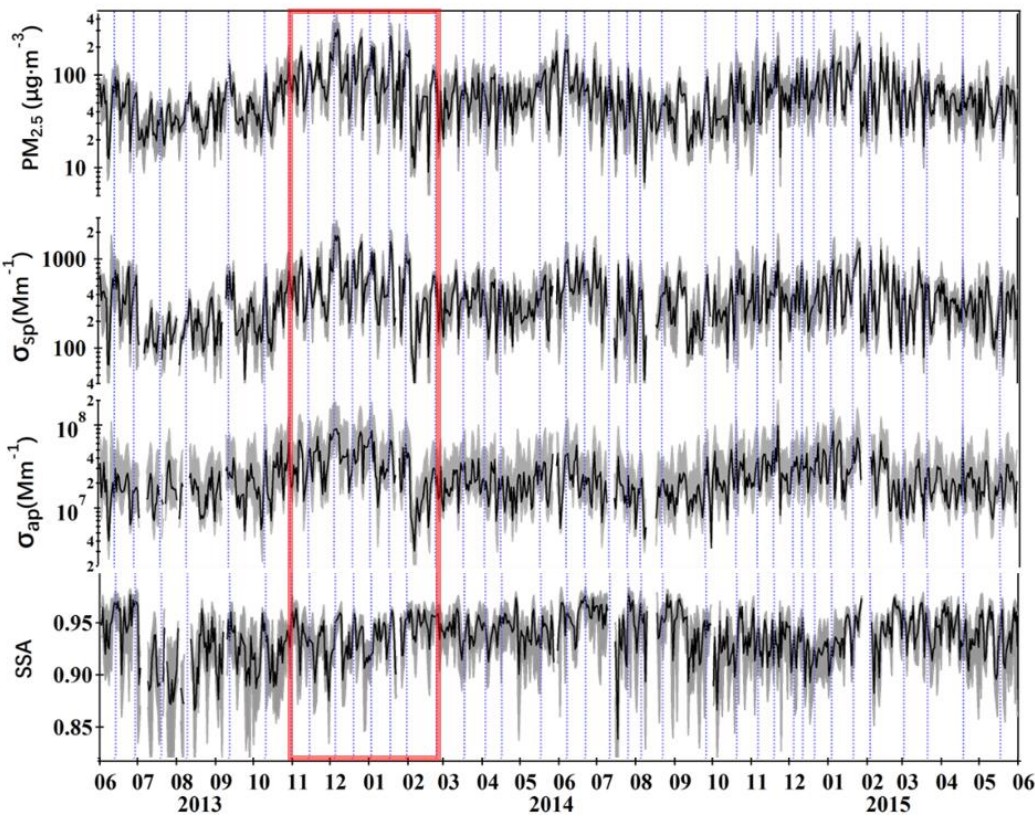

Figure 1. Daily averaged PM$_{2.5}$ concentations, scattering coefficients ($\sigma_{sp}$, $\lambda$ = 525 nm), absorption
coefficients $\sigma_{ap}$, $\lambda$ = 520 nm), and single-scattering albedo (SSA, $\lambda$ = 520 nm) at SORPES in June
2013 - May 2015. The grey shaded areas show the 10[th] to 90[th] percentile range of hourly-averaged
data. The blue vertical line denote pollution episodes.The red box shows the period that is analyzed
in more detail in Figure 8.



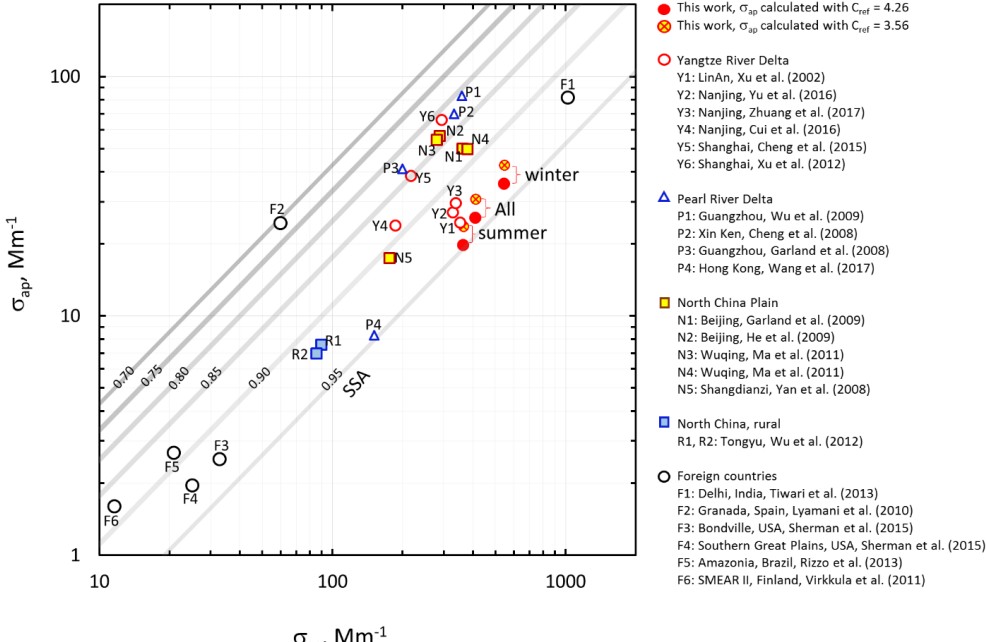

Figure 2. Average absorption coefficient ($\sigma_{ap}$) vs. average scattering coefficient ($\sigma_{sp}$) of green light at SORPES (this work) and selected Chinese and foreign sites. For SORPES the averages for the whole period (Table 1) and for summer and winter (Table 3) are presented with two symbols: the solid red circles and the circles with the red cross for which $\sigma_{ap}$ was calculated by using $C_{ref} = 4.26$ and 3.56, respectively. For details of measurement methods, periods, wavelengths and references of the comparison sites see Table 2. Constant single-scattering albedo (SSA) is depicted by the solid grey lines.





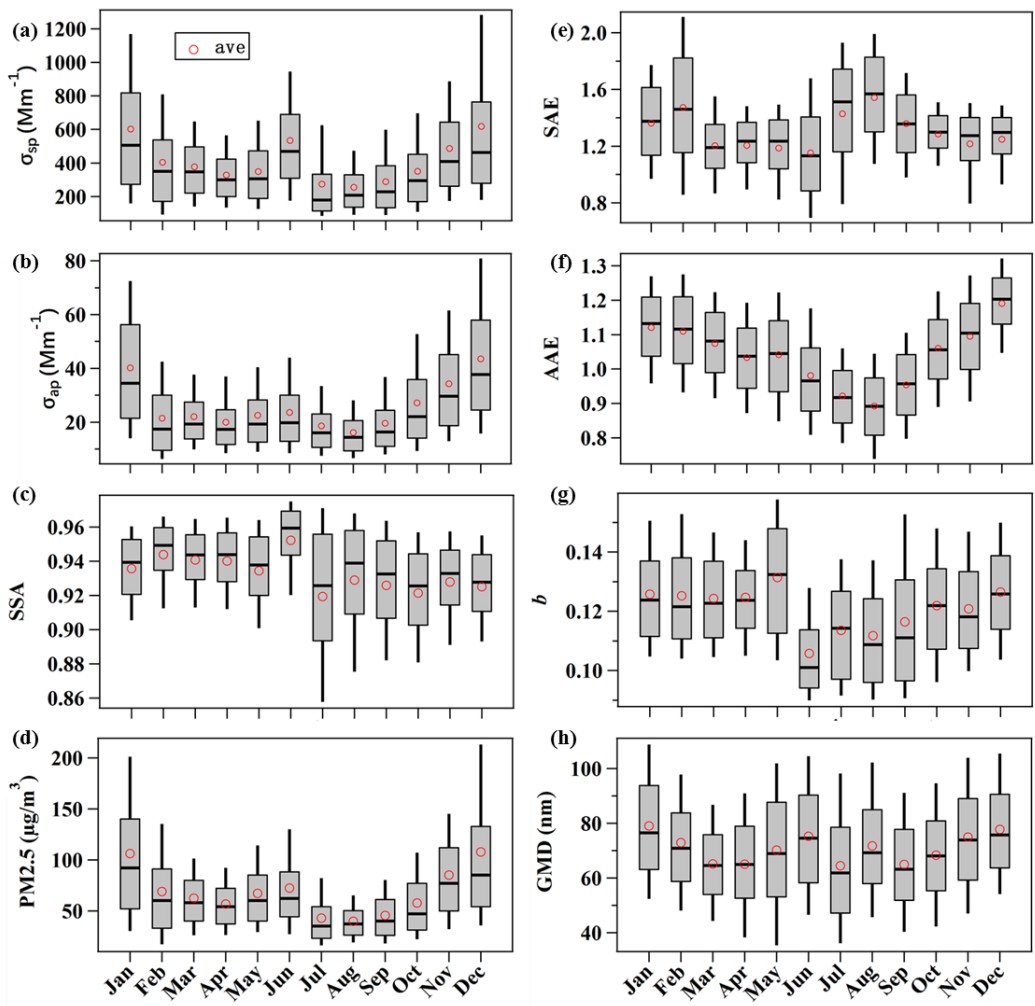

Figure 3. Seasonal cycle of selected aerosol optical properties and supporting parameters at SORPES. a) Scattering coefficient at 525 nm, b) absorption coefficient at 520 nm, c) backscatter fraction at 525 nm, d) single-scattering albedo at 520 nm, e) scattering Ångström exponent between 450 and 635nm, f) absorption Ångström exponent between 370 and 950nm, g) PM$_{2.5}$ concentration and h)geometric mean diameter for 6-800nm particles.The solid lines represent median values, red cycles stand for averages, the boxes are 25$^{th}$ and 75$^{th}$ percentiles and the whiskers represent the 10$^{th}$ and 90$^{th}$ percentiles.





Figure 4. Diurnal variations of a) scattering coefficient ($\sigma_{sp}$) at $\lambda = 525$ nm and absorption coefficient ($\sigma_{ap}$) at $\lambda = 520$ nm, b) Ångström exponents of scattering (SAE) and absorption (AAE), c) single-scattering albedo (SSA) at $\lambda = 520$ nm, d) backscatter fraction ($b$) at $\lambda = 525$ nm, e) photochemical age of air mass ($-\log(NO_x/NO_y)$, and f) particle number size distribution and the geometric mean diameter (GMD) in the four seasons. The values are the averages of the corresponding hour in the four seasons coded by the colors.





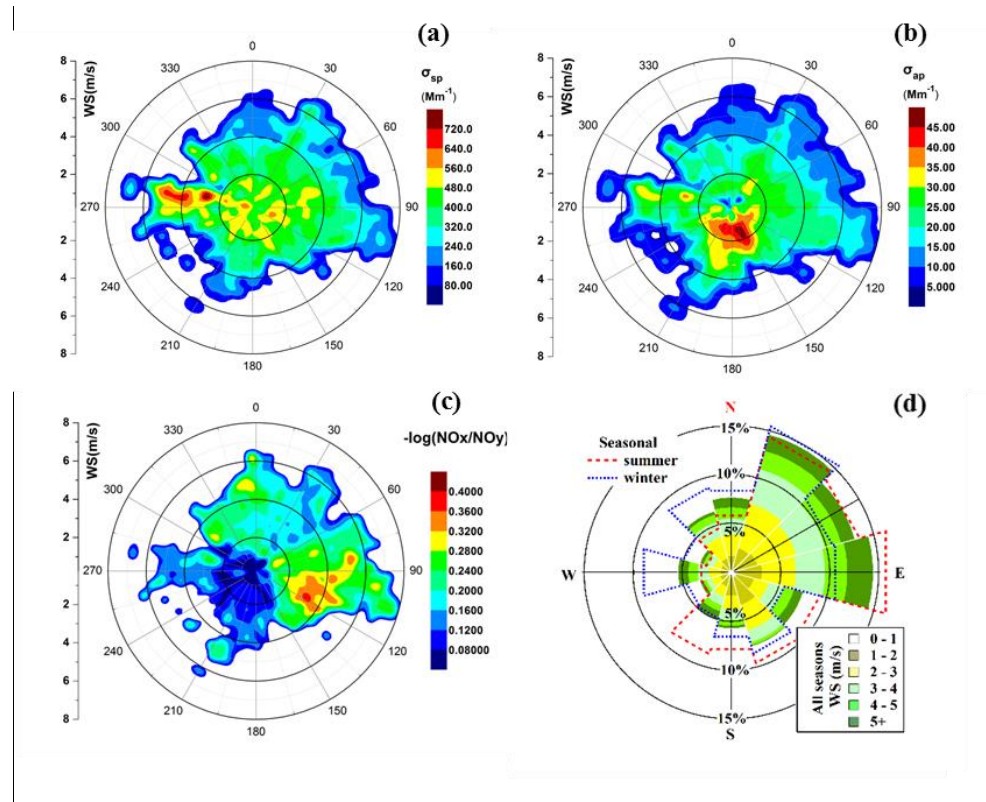

Figure 5. Counter polar plot for (a) Scattering coefficient at 525 nm (b) Absorption coefficient at 520nm (c) photochemical age = $-\log (NO_x/NO_y)$ (d) wind rose for 2-years.



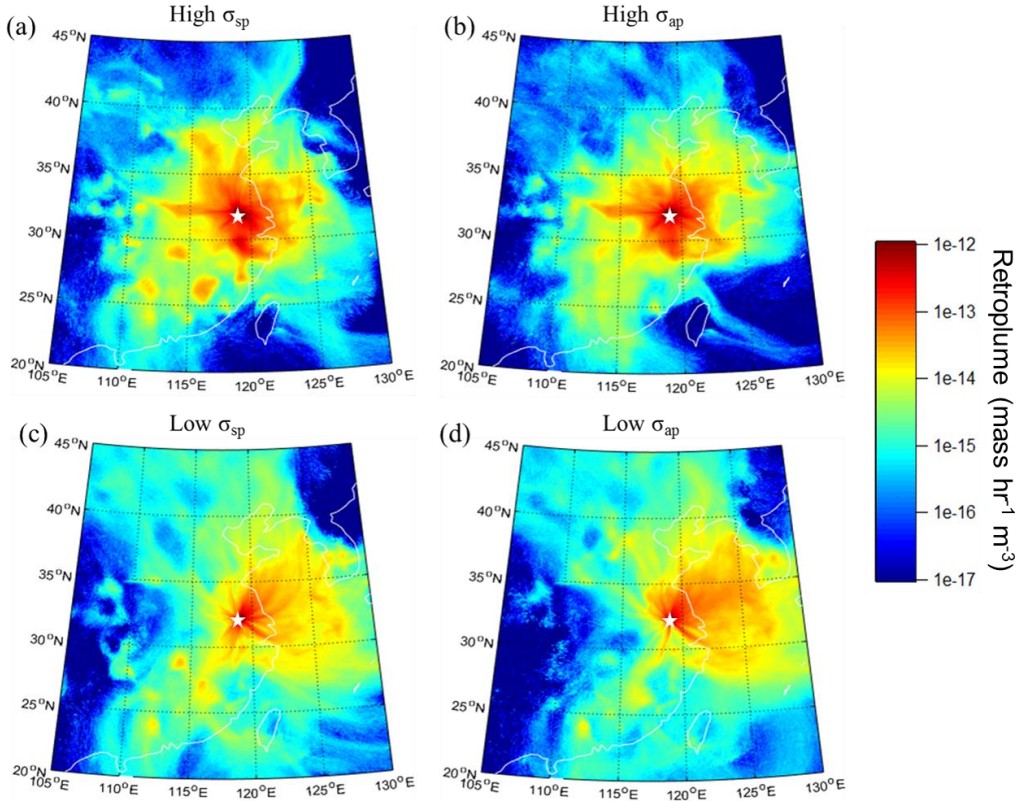

Figure 6. Averaged 72-h retroplumes for the highest 10% of daily averages of a) $\sigma_{sp}$ and b) $\sigma_{ap}$ of each month and for the lowest 10% of daily averages of c) $\sigma_{sp}$ and d) $\sigma_{ap}$ of each month.



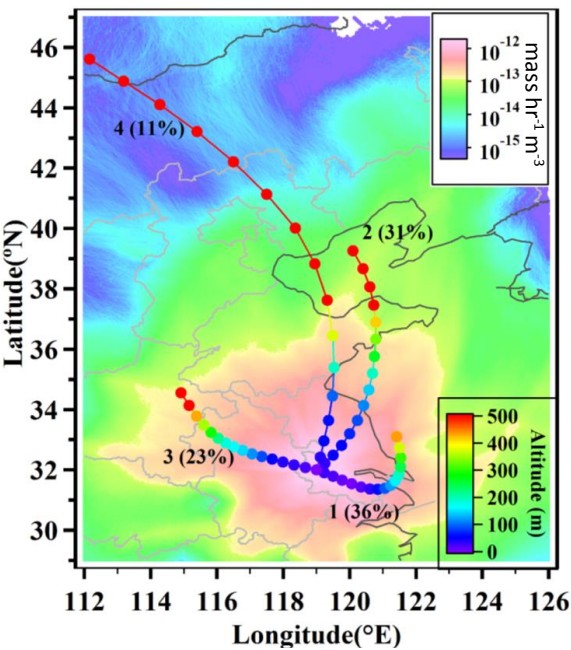

Figure 7. The average backtrajectories of the clusters obtained from the backtrajectory cluster analysis for the winter period 2013/11/01 – 2014/02/28. The length of the trajectories is 48 hours. The average 72-h retroplume for the same period is depicted with the background color.



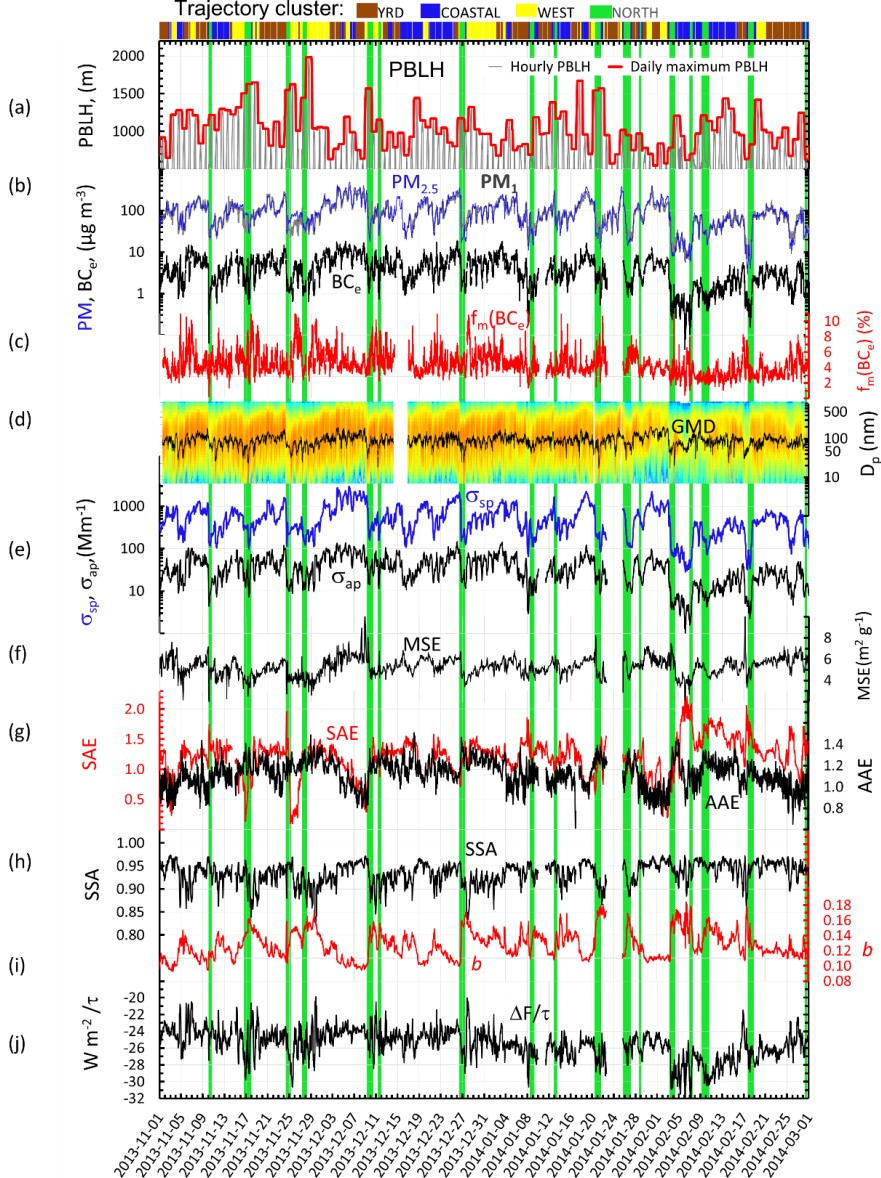

Figure 8. Hourly-averaged aerosol optical properties, mass concentrations and modeled boundary layer height during a polluted 4-month period 2013/11/01 – 2014/02/28. Top: time series of trajectory clusters shown with color coding, a) PBLH: planetary boundary layer height, b) PM₂.₅ and PM₁: Mass concentrations of particles smaller than 2.5 µm and 1 µm, BCₑ: equivalent black carbon concentration, c) fₘ(BCₑ): mass fraction of BCₑ, d) particle number size distribution and GMD: geometric mean diameter, e) σ_sp: scattering coefficient at λ=525 nm, σ_ap: absorption coefficient at λ=520 nm, f) MSE: mass scattering efficiency at λ =525 nm, g) SAE and AAE: scattering and absorption Ångström exponents, h) SSA: single-scattering albedo at λ = 520 nm, i) b: backscatter fraction at λ =525 nm, and j) ΔF/τ: aerosol radiative forcing efficiency at λ = 525 nm.





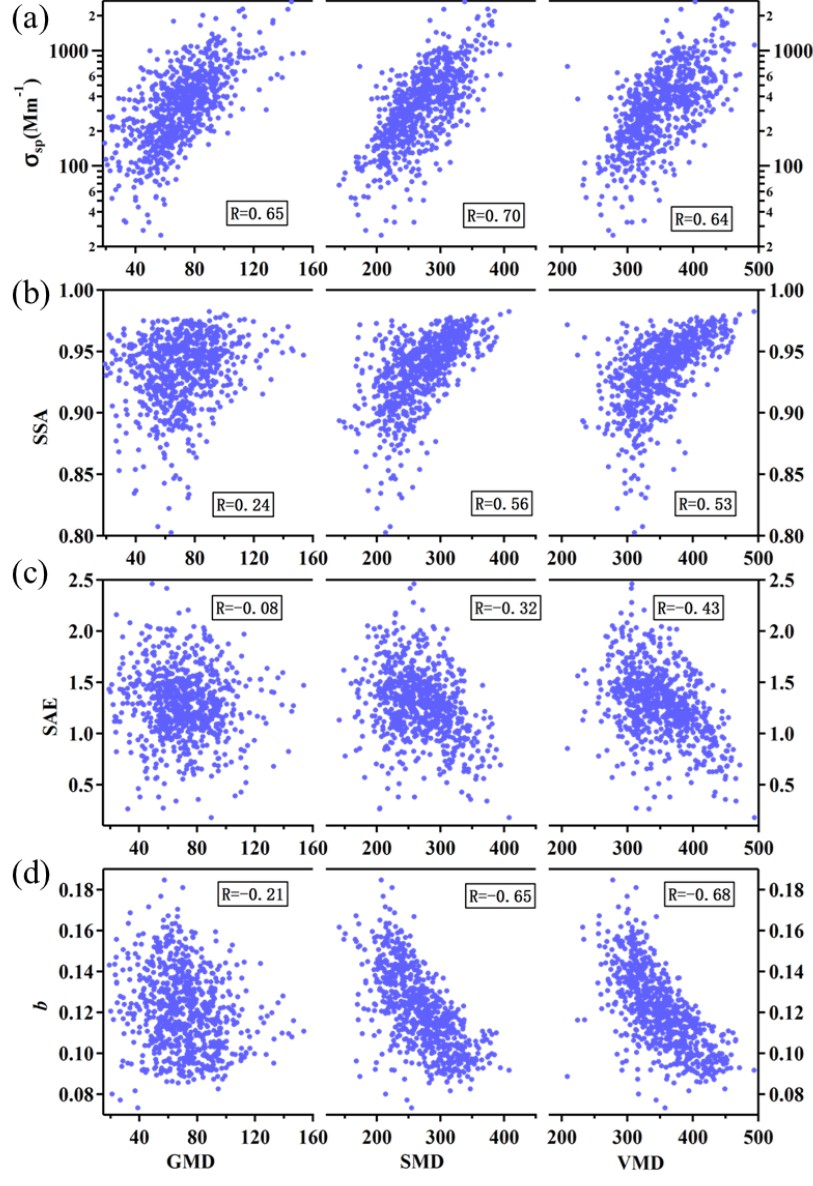

Figure 9. Relationships between a) scattering coefficient at $\lambda = 525$ nm, b) SSA at $\lambda = 525$ nm, c) Ångström exponent of scattering (SAE), and d) backscatter fraction at $\lambda = 525$ nm and the effective mean diameters GMD, SMD and VMD. The correlation coefficients are those obtained from linear regressions.



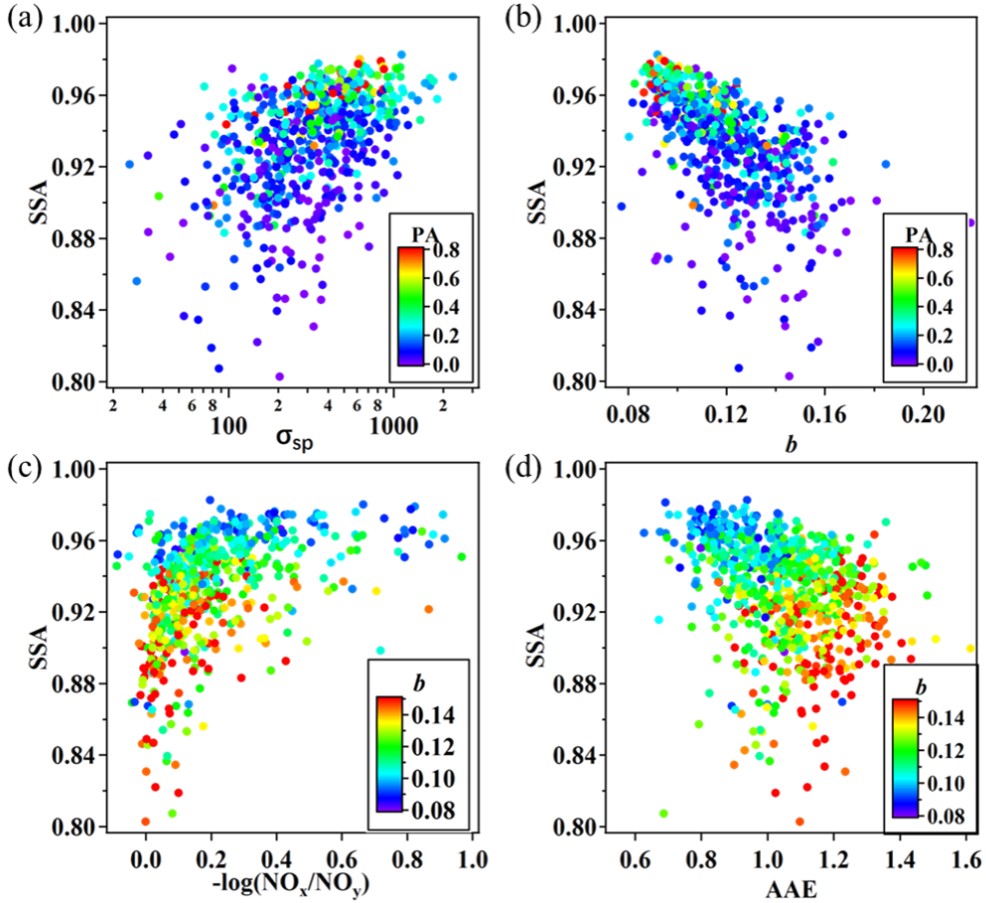

Figure 10. Relationships of single-scattering albedo SSA at $\lambda = 525$ with a) scattering coefficient, b) backscatter fraction, c) photochemical age and d) Ångström exponent of absorption. In a) and b) the data points are colorcoded with the photochemical age (PA) and in c) and d) with backscatter fraction.





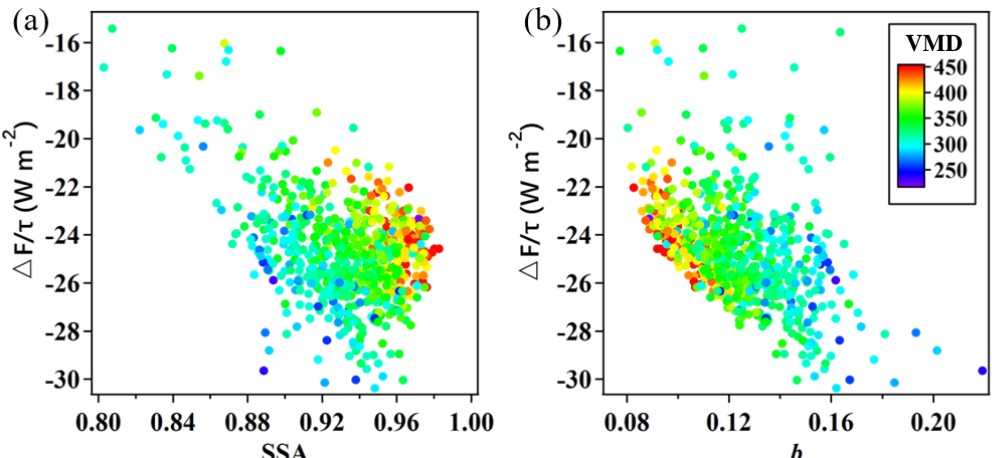

Figure 11. Radiative forcing efficiency ($\Delta F/\tau$) at $\lambda = 525$ nm as a function of a) SSA and b) backscatter fraction and volume mean diameter (VMD in nm) of the size distribution.

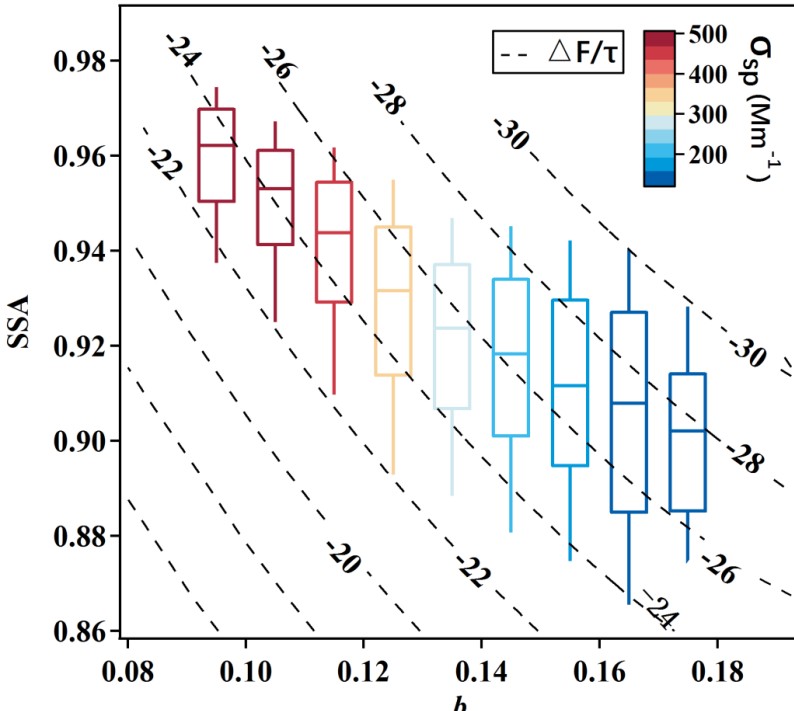

Figure 12. Relationship of SSA with the backscatter fraction and the radiative forcing efficiency at $\lambda = 525$ nm depicted with the isolines in W m$^{-2}$ $\tau^{-1}$.



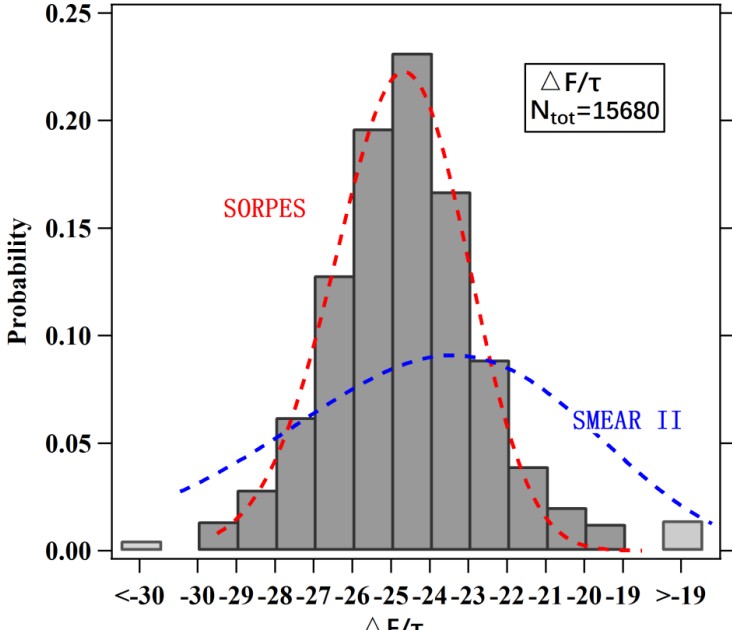

Figure 13. Probability distribution of the radiative forcing efficiency at SORPES and at SMEAR II
5   (Virkkula et al., 2011).





**TABLES**

Table 1. Statistical summary of aerosol optical properties and associated supporting parameters
5  measured at SORPES, Nanjing, in July 2013 – May 2015. $N/N_{tot}$: fraction of hours with valid
observations.

| | $N/N_{tot}$ | $\lambda$ | | percentiles | | | | | | |
| --- | --- | --- | --- | --- | --- | --- | --- | --- | --- | --- |
| | (%) | (nm) | ave ± std | 1% | 10% | 25% | 50% | 75% | 90% | 99% |
| $\sigma_{ap}$ (Mm$^{-1}$) | 91.2 | 370 | 39 ± 30 | 5.9 | 13 | 19 | 31 | 50 | 79 | 144 |
| | | 520 | 26 ± 19 | 4.2 | 8.8 | 13 | 21 | 33 | 51 | 51 |
| | | 880 | 16 ± 11 | 2.6 | 5.4 | 8.0 | 13 | 20 | 31 | 31 |
| $\sigma_{sp}$ (Mm$^{-1}$) | 95 | 635 | 330 ± 260 | 36 | 87 | 143 | 258 | 426 | 642 | 1326 |
| | | 525 | 410 ± 320 | 51 | 116 | 188 | 332 | 531 | 798 | 1594 |
| | | 450 | 490 ± 370 | 65 | 145 | 234 | 406 | 634 | 941 | 1857 |
| SSA | 91.2 | 370 | 0.94 ± 0.03 | 0.84 | 0.89 | 0.92 | 0.94 | 0.96 | 0.97 | 0.98 |
| | | 520 | 0.93 ± 0.03 | 0.83 | 0.89 | 0.92 | 0.94 | 0.96 | 0.97 | 0.98 |
| | | 880 | 0.92 ± 0.04 | 0.78 | 0.87 | 0.90 | 0.93 | 0.95 | 0.96 | 0.98 |
| AAE | 91.2 | 370/950 | 1.04 ± 0.15 | 0.72 | 0.84 | 0.93 | 1.04 | 1.16 | 1.24 | 1.38 |
| | | 370/880 | 1.06 ± 0.18 | 0.71 | 0.83 | 0.93 | 1.06 | 1.19 | 1.30 | 1.52 |
| | | 470/660 | 0.96 ± 0.18 | 0.60 | 0.73 | 0.83 | 0.95 | 1.09 | 1.20 | 1.39 |
| SAE | 95 | 450/635 | 1.30 ± 0.34 | 0.40 | 0.88 | 1.11 | 1.30 | 1.49 | 1.73 | 2.14 |
| | | 450/525 | 1.31 ± 0.37 | 0.37 | 0.87 | 1.10 | 1.29 | 1.51 | 1.80 | 2.23 |
| | | 525/635 | 1.28 ± 0.32 | 0.41 | 0.87 | 1.10 | 1.30 | 1.48 | 1.67 | 2.04 |
| $b$ | 95 | 450 | 0.11 ± 0.02 | 0.08 | 0.09 | 0.10 | 0.11 | 0.12 | 0.14 | 0.16 |
| | 95 | 525 | 0.12 ± 0.02 | 0.08 | 0.09 | 0.11 | 0.12 | 0.13 | 0.16 | 0.17 |
| | 95 | 635 | 0.14 ± 0.03 | 0.09 | 0.10 | 0.12 | 0.14 | 0.15 | 0.17 | 0.20 |
| $\triangle F/\tau$ (W/m$^2$) | 91.2 | 525 | -24.6 ± 2.2 | -29.5 | -27.1 | -25.9 | -24.7 | -23.5 | -22.1 | -17.2 |
| PM$_{2.5}$ (μg/m$^3$) | 98.9 | \ | 68 ± 50 | 10 | 22 | 34 | 56 | 85 | 126 | 295 |
| GMD (nm) | 92.9 | \ | 71 ± 22 | 24 | 44 | 56 | 69 | 84 | 99 | 132 |
| VMD (nm) | 92.9 | \ | 350 ± 46 | 254 | 294 | 316 | 346 | 383 | 416 | 455 |





Table 2. Published scattering coefficients (σ_sp), absorption coefficients (σ_ap) and single-scattering albedos (SSA) at selected Chinese and foreign sites. The numbers in parentheses show the wavelength of the respective parameter in nm.

| | location | coordinates | period | σ_sp (λ) Mm⁻¹ (nm) | σ_ap (λ) Mm⁻¹ (nm) | SSA (λ) - (nm) | instruments scattering | absorption | reference |
|---|---|---|---|---|---|---|---|---|---|
| YRD | Nanjing | 32.2°N, 118.7°E | 2011/3-2011/4 | 329(550) | 28.1(532) | 0.89(532) | TSI3563 | PASS | Yu et al., 2016 |
| | Nanjing | 32.05°N, 118.78°E | 2014/3-2016/2 | 338(550) | 29.6(550) | 0.901(550) | Aurora 3000 | AE31 | Zhuang et al.,2017 |
| | Linan | 31.30°N, 119.73°E | 1999/11 | 353(530) | 23(565) | 0.93 | Radiance Research | PSAP | Xu et al., 2002 |
| | Nanjing | 32.2°N, 118.7°E | 2012/8-2012/9 | 186(532) | 23.9(532) | 0.88(532) | PASS | PASS | Cui et al.,2016 |
| | Shanghai | 31°18'N, 121°29'E | 2010/12-2012/10 | 217(525) | 38(532) | 0.83(532) | Aurora-1000 | AE31 | Cheng et al., 2015 |
| | Shanghai | 31°18'N, 121°29'E | 2010/12-2011/3 | 293(525) | 66(532) | 0.81(532) | M9003 | AE31 | Xu et al., 2012 |
| PRD | Guangzhou | 23°00'N, 113°21'E | 2004-2007 | 358(525) | 82(532) | 0.81(525) | M9003 | AE31 | Wu et al., 2009 |
| | XinKen | 22.6°N 113.6°E | 2004/10 | 333(550) | 70(550) | 0.83(550) | TSI3563 | MAAP | Cheng et al., 2008 |
| | Hongkong | 22.22°N, 114.25°E | 2012/2-2015/2 | 151(550) | 8.3(550) | 0.93(550) | TSI3563 | AE31 | Wang et al., 2017 |
| | Guangzhou | 23.55°N, 113.07° E | 2006/7 | 200(550) | 42.5(532) | 0.83(532) | TSI3563 | PAS | Garland et al., 2008 |
| Northern China | Shangdianzi | 40°39'N, 117°07' E | 2013/4-2015/1 | 174.6(525) | 17.5(532) | 0.88(525) | M9003 | AE31 | Yan et al., 2008 |
| | Beijing | 39.51°N, 116.31°E | 2006/8 | 361(550) | 51.8(532) | 0.86(532) | TSI3563 | PAS | Garland et al., 2009 |
| | Beijing | 39°59'N, 116°19' E | 2005/1-2006/12 | 288(525) | 56(525) | 0.80(525) | M9003 | AE16 | He et al., 2009 |
| | Tongyu | 44.56°N, 122.92°N | 2010/spring | 89.22(520) | 7.61(520) | 0.90(520) | Aurora-3000 | AE31 | Wu et al., 2012 |
| | | | 2011/spring | 85.34(520) | 7.01(520) | 0.90(520) | | | |
| | Wuqing | 39.3°N, 117.0°N | 2009/spring | 280(550) | 47(637) | 0.82(637) | TSI3563 | MAAP | Ma et al., 2011 |
| | | | 2009/summer | 379(550) | 43(637) | 0.86(637) | | | |
| World wide | Delhi, IND | 28° 37'N, 77°12'E | 2011/12-2012/3 | 1027(500) | 86(500) | 0.93(500) | TSI3563 | AE31 | Tiwari et al., 2015 |
| | Amazonia, BRA | 2°36'S, 60°13' E | 2008/02-2011/02 | 21(550) | 2.3(637) | 0.86(637) | TSI3563 | MAAP | Rizzo et al., 2013 |
| | BND, USA | 40.0°N, 88.4°E | 2010-2013 | 32.9(550) | 2.51(550) | 0.917(550) | TSI3563 | PSAP | Sherman et al., 2015 |
| | SGP, USA | 36.6°N, 97.5°W | 2010-2013 | 25.1(550) | 1.95(550) | 0.913(550) | TSI3563 | PSAP | |
| | Granada, ESP | 37.16°N,3.58°W | 2005/12-2007/11 | 60(550) | 21(637) | 0.68(637) | TSI3563 | MAAP | Lyamani et al., 2010 |
| | SMR II, FIN | 61°51'N, 24°18'E | 2006/10-2009/05 | 18(550) | 2.1(550) | 0.86(550) | TSI3563 | AE31 | Virkkula et al., 2011 |



Table 3. Statistical summary of hourly-averaged aerosol optics data and associated parameters in summer (June–August), autumn (September–November), winter (December–February), and spring (March–May).

| | $\lambda$ | JJA | | SON | | DJF | | MAM | |
|---|---|---|---|---|---|---|---|---|---|
| | (nm) | ave±std | median | ave±std | median | ave±std | median | ave±std | median |
| $\sigma_{ap}$ (Mm$^{-1}$) | 370 | 29±21 | 24 | 41±29 | 32 | 56±37 | 47 | 32±19 | 28 |
| | 520 | 20±13 | 17 | 27±18 | 22 | 36±24 | 30 | 22±13 | 19 |
| | 880 | 13±8 | 11 | 16±11 | 13 | 21±14 | 17 | 13±8 | 11 |
| $\sigma_{sp}$ (Mm$^{-1}$) | 635 | 290±250 | 206 | 298±218 | 245 | 429±350 | 326 | 284±164 | 251 |
| | 525 | 364±295 | 274 | 378±273 | 313 | 545±425 | 427 | 351±196 | 317 |
| | 450 | 433±333 | 342 | 457±324 | 379 | 654±486 | 529 | 421±228 | 386 |
| SSA | 370 | 0.94±0.03 | 0.95 | 0.93±0.03 | 0.93 | 0.93±0.03 | 0.94 | 0.94±0.03 | 0.94 |
| | 520 | 0.94±0.04 | 0.95 | 0.93±0.03 | 0.93 | 0.93±0.02 | 0.94 | 0.94±0.02 | 0.94 |
| | 880 | 0.91±0.06 | 0.93 | 0.91±0.04 | 0.92 | 0.92±0.03 | 0.93 | 0.93±0.03 | 0.93 |
| AAE | 370/950 | 0.94±0.14 | 0.93 | 1.04±0.14 | 1.04 | 1.14±0.13 | 1.16 | 1.05±0.13 | 1.06 |
| | 370/880 | 0.95±0.16 | 0.93 | 1.06±0.18 | 1.05 | 1.17±0.16 | 1.18 | 1.06±0.16 | 1.06 |
| | 470/660 | 0.84±0.16 | 0.83 | 0.96±0.17 | 0.95 | 1.07±0.16 | 1.08 | 0.97±0.16 | 0.97 |
| SAE | 450/635 | 1.36±0.43 | 1.39 | 1.28±0.27 | 1.30 | 1.36±0.36 | 1.34 | 1.20±0.27 | 1.22 |
| | 450/525 | 1.41±0.45 | 1.42 | 1.29±0.28 | 1.30 | 1.39±0.39 | 1.35 | 1.17±0.29 | 1.19 |
| | 525/635 | 1.31±0.41 | 1.35 | 1.28±0.26 | 1.30 | 1.31±0.34 | 1.33 | 1.23±0.26 | 1.26 |
| $b$ | 450 | 0.11±0.02 | 0.10 | 0.12±0.02 | 0.11 | 0.12±0.01 | 0.12 | 0.12±0.01 | 0.12 |
| | 525 | 0.11±0.02 | 0.11 | 0.12±0.02 | 0.12 | 0.13±0.02 | 0.12 | 0.13±0.02 | 0.13 |
| | 635 | 0.12±0.03 | 0.11 | 0.13±0.03 | 0.13 | 0.14±0.02 | 0.14 | 0.15±0.02 | 0.15 |
| $\triangle F/\tau$ (W/m$^2$) | 525 | -23.3±2.5 | -23.8 | -23.8±2.1 | -23.9 | -25.3±1.7 | -25.1 | -25.7±1.6 | -25.7 |
| PM$_{2.5}$(μg/m$^3$) | \ | 51±36 | 43 | 63±41 | 54 | 95±69 | 78 | 62±33 | 57 |
| GMD (nm) | \ | 71±24 | 68 | 70±21 | 68 | 77±22 | 74 | 67±21 | 66 |
| VMD (nm) | \ | 364±54 | 363 | 347±41 | 346 | 350±46 | 346 | 342±43 | 337 |

