# Peer review of "Aerosol Optical Properties at SORPES in Nanjing, East China"

_Atmospheric Chemistry and Physics, 2017_

## Referee Comment (RC1) · Anonymous Referee #1 · 7 Nov 2017

The article entitled, "Aerosol optical properties at SORPES in Nanjing, East China" focuses on the analysis of a long-term (2 year) dataset of aerosol measurements at the SORPES site. The article does present a new dataset that assists in characterizing the pollution at this site; in addition, the authors have compared some of the findings to other sites. The methods are well outlined. I do believe there are some assumptions that do need more detail or explanation. This includes the correction done for the malfunctioning heater in the nephelometer, as well as the assumption of the large role of new particle formation. I have expanded on both of these in the specific comments below. In addition, I would recommend that the conclusions are expanded a little to discuss the relevance and importance of the dataset from this site now that the pollution is characterized in this pollution. For example, in relation to other long-term sites in the

area and in China in general, what role can this site play?

I would recommend that the article is accepted after the comments below are considered.

Specific comments

Section 2.1. I believe a map of Nanjing and surrounding area, including its location relative to other cities in China would be helpful in understanding the site, and also in the analysis of the back trajectories and wind roses.

Page 5. How were the data points that were impacted by the faulty heater in the nephelometer identified? Also, what tests were done to ensure that this correction is not creating an artefact? For example, were those points across the monitoring period or in specific time period? How sensitive are your results to this correction (e.g. both the fact that it had to be done and the constants c and g that were used)? I understand that this correction was done in order to use the data, and that makes sense. However, I would recommend adding some more information on the impact of this assumption on the results.

Page 5 line 34, how was it decided to use the wavelength insensitive Cref? How was it determined that this was the most appropriate?

Page 7 line 26, the Wang et al reference for using a density of 1.7 g/cm-3 is not local to this monitoring station. How was it determined that it is applicable to this dataset?

Page 8 line 6, the first sentence in the first paragraph in 2.4 is not complete.

Page 8, line 10, I would recommend defining what is considered a "small" and "high" value.

Page 10, paragraph starting on line 25 details the AAE range. It is stated here that because the wavelength independent Cref was used, these values are smaller than if a wavelength dependent Cref was used. In the conclusions, it is acknowledged that

using a different Cref would lead to the conclusion that there is brown carbon. As this is an uncertainty that has an impact on the conclusion, I would recommend discussing this on page 10 as well.

Page 11, section 3.2. In the reasons for the seasonal variation there is no mention of the possibility of a seasonal dependence of emissions impacting on the seasonal cycle. Why is that?

Page 12, line 8-10. The increase in SSA is attributed to new particle formation (NPF). However, transport of scattering particles (which is seen in the later analysis) could also lead to a peak in the afternoon when the aerosols are well-mixed. It is not clear why this was only attributed to new particle formation. This assumption that NPF events are responsible for this then also plays a role in the assessment that the aerosols responsible for higher scattering originated outside of Nanjing (page 15). On page 15, it is stated that this is only true for NPF events, however this caveat is not stated in conclusions when this point is made again (page 22, line 21-22). On page 18, line 7 it is stated that there was a previous study looking at the dominance of NPF at the site. This is a key point that should be moved further up in the text to provide support to the discussion of NRF starting on page 12.

Page 14, line 22, the dilution with higher wind speeds is attributed to vertical mixing specifically, but I would imagine that horizontal transport and increased dispersion with high wind speeds would also play a role.

Page 15, line 20-21, what sources of emissions are in direction where the fresh traffic emissions were measured? If there are not obvious roads there, then the assessment of these as fresh traffic emissions would be incorrect.

Page 15, line 23-25, the WNW sector was said to have very few points (page 14, line 25-26), thus how robust is the assessment of the age that comes up with controversial results? It does look from Figure 5 that the winds come from this direction mostly in winter, can that help to understand the potential source of pollution (e.g. NOx not

emitted with BC)?

Section 3.6.1 states that AOP are not as closely related to GMD as SMD and VMD. Then why was GMD used in Figure 8 and Figure 4, where the general relationships and trends are discussed?

Technical corrections

Page 1 line 26, "in" is duplicated

Page 1 line 32 "of" is duplicated

Page 1 line 34, "b" is referred to but is not yet defined.

Page 2 line 26, as stated, there are many studies in these areas, and thus the ones listed are only some of the studies. I would recommend using "e.g." before the references to indicate this.

Page 2 line 35, what year was SORPES station started?

Page 4 line 2, "hygroscopic growth is usually significant when RH increase above 50%RH" is this statement specific to this site or a general statement?

Page 4, line 27, For readers who are not familiar with SORPES, I would recommend moving this reference for the overview of the site (Ding et al 2013 and 2016) at the beginning of this section.

Section 2.3.4, the terms for equations 9-11 are not all defined in the text.

Page 8, line 34, PM2.5 are particles with an aerodynamic diameter less than or equal to 2.5 um.

Page 12, line 5, "wee hours" is colloquial; I would recommend using another term.

---

## Referee Comment (RC2) · Anonymous Referee #2 · 19 Nov 2017

This manuscript presents a summary and analysis of 2 years of aerosol optical measurements at a monitoring site near Nanjing, China in the Yangtze River Delta region. The measurements are extremely interesting and represent a compilation of some of the most complete, co-measured aerosol optical properties (AOP) data available in eastern China. These AOP measurements are complemented by particle size distribution and meteorological observations. The authors do a nice job of summarizing the data and analyzing it to describe the radiative forcing consequences of the heavy aerosol loading present in this area.

Some improvements need to be made to the manuscript prior to publication. These go beyond minor and typographical corrections and so should be placed in the "major revision" category, but I do not wish to imply that this work is substantially flawed. It

could just be a better with a few changes.

My major scientific concerns are:

1) The heater in the nephelometer did not work for much of the 2-year observations period. Given the warm temperatures and high humidity during much of the year and the operation of the instruments in a (presumably cooler) indoor laboratory, there may be considerable aerosol water remaining even if the data are restricted to calculated RH <50%. The authors have not attempted to compare the SMPS size distribution measurements with the nephelometer by using Mie theory to calculate ambient scattering. There should be a short section that compares all measurements to whatever extent possible, including scattering, backscatter fraction, and aerosol mass. In other words, show internal consitency for the measurements analyzed. Then the reader would be more assured that the instruments are operating within their expected uncertainty.

2) Measurement variability is often discussed as a mean and standard deviation (e.g., in the abstract). However, these metrics are not descriptive when the parameter is not normally distributed, as is the case for most of the extensive parameters. In time plots, the data are shown to vary on a logarithmic scale. Thus it would be more logical to calculate, display, and discuss geometric means and geometric standard deviations. I appreciate that the 10th and 90th percentile values are also used as an indicator of variability; this is good and should be retained.

3) The time plots (Figs 1 and 8) are not very useful because there are so many parameters displayed. Figures 1 and 8 in particular are extremely dense, and it is very hard to see systematic changes during the "episodes" that are discussed in the text. I recommend focusing on one exemplary "episode" for Figure 8 and eliminating the other times shown on this graph. Then average values, described in a table, could be shown to describe the remaining episodes, e.g. length of each episode, changes in optical properties from episode start to finish, etc. This would be more quantitative and useful to the reader. Please make a quantitative definition of an "episode" and use it

consistently to identify these events.

4) The authors may wish to consider explicitly if biomass burning (mostly agricultural) could explain the cases of very high scattering but low absorption coming from the west-northwest direction.

5) On p. 14, line 30, an assumption is made that new particle formation and growth to optically significant sizes are necessary to explain increases in aerosol scattering. In fact, light scattering is roughly proportional to particle volume or mass because the scattering efficiency increases very approximately linearly across the broad accumulation mode, leaving scattering proportional to diameter cubed. Most of the increase in mass, hence light scattering, is almost certainly occurring on this accumulation mode. The growth of recently formed particles is very unlikely to compete with condensational growth on existing accumulation mode particles because of this strong diameter dependence of scattering.

6) In Section 3.6.2 it is confusing to describe the radiative effect as "decreasing" or "getting smaller" as light scattering increases. The magnitude of the effect is increasing; the sign of the effect is negative. I recommend explicitly saying that the magnitude of the RFE is increasing and use this consistently throughout the text.

Technical comments:

1) P1, line 34. The parameter "b" is not defined.

2) P2, line 15. Is the acronym "LAC" ever used again? There is no need to define an acronym that is not used later.

3) P3, line 7. Change "month" to "months".

4) P10, line 23. Change "there" to "their"

5) P16, line 23. Correct "to1".

6) Fig. 1. The choices of gray-scale error bands make the whole graph look out-of-

focus. Is there some better way to do this?

7) Fig. 1. I don't see any "blue vertical lines". It's not at all clear how the vertical lines that are shown on the graph are related to any "episodes".

8) Fig. 1. The scales/units for the absorption coefficient are incorrect.

9) Fig. 4. What is the color scale in Fig. 4f? Is this dN/dlog(Dp)? Is this an average (mean) sized distribution over the entire period of the observations? Is it rational to average size distributions in this way? For example, an accumulation-mode-dominated size distribution averaged with one dominated by an Aitken mode peak might result in a size distribution that represents neither case. Would it be better to show temporal evolution of modal parameters (e.g., accumulation mode Dgeom, Aitken/nuclei mode Dgeom) calculated and displayed as in Fig a-e?

10) Fig. 4. Can you show some sparsely plotted error bars that represent the 10th and 90th percentile values on the plots? This might make the graphs too crowded, but there needs to be some way to show if the changes in the parameters are significant relative to the hourly variability of the data.

11) Fig. 5. Please explain these color contour plots. What is a "counter polar plot". What do the colors represent?

12) Fig. 6. Please explain the color scale and the units. What does "retroplume" as a value mean exactly?

13) In Fig. 7 the trajectories are labeled "1", "2", etc. In Fig. 8 they are given names. Please use the names instead of numbers in Fig. 7.

14) The color scale for Fig. 12 is readable by most color-blind people, while the color scales on the other figures are not. Please use this color scale on all figures for the ∼7% of readers who are color-blind.

15) Table 1. Show geometric means and standard deviations.

16) Table 2. Add the current results to this table so we can compare directly.

---

## Author Comment (AC1) · 1 Mar 2018

The comment was uploaded in the form of a supplement:
https://www.atmos-chem-phys-discuss.net/acp-2017-863/acp-2017-863-AC1-supplement.pdf

---

## Author Response (AR1)

**Shen et al.: Aerosol Optical Properties at SORPES in Nanjing, East China Atmos. Chem. Phys. Discuss., https://doi.org/10.5194/acp-2017-863**

**Replies to the reviewer's comments**

First of all, the authors thank the reviewers for their evaluations, they helped improving the paper. The text was corrected according to most of the suggestions of the reviewers. The largest changes were

1) We now emphasize that particles formed by NPF do not have a significant effect on aerosol optical propoerties during the day they are formed

2) We added an evaluation of the effect of using wavelength-dependent $C_{ref}$ in the calculation of absorption coefficients and Ångström exponents of absorption (AAE) and the consequences for the interpretation of the AAE. Related to this, in Tables 1 and 2 we replaced one line of AAE values, that were originally calculated using the wavelength pair 370/880 to 470/950 since the latter is used in the aethalometer model by Zotter et al (2017) so it was considered to be more relevant.

3) We evaluated different hygroscopicity correction parameterizations and chose to use the one presented by Carrico et al. (2003). Evaluations are presented in the supplement. Consequently the scattering coefficients presented in the results and tables changed but not so much that any of the conclusions would change.

4) We present also a comparison of Mie-modeled scattering and backscattering coefficients with the measured ones and a comparison of scattering coefficient and PM2.5 mass concentration in the supplement.

5) We added one figure that shows the diurnal cycle of size distributions during the whole period and during different NPF event class days together with the SSA in these same classes. This was considered to be relevant in view of the discussion of the effect of NPF on AOPs.

6) A previously undiscovered malfunction of the DMPS was found so the fraction of good size distribution data decreased from 92.9% to 77.8%. This affected mainly the descriptive statistics in Tables 1 and 3 but does not affect any of the conclusions.

In the revised text major changes and additions are highlighted with yellow. Minor language corrections have not been indicated.

Below the **reviewers' comments are written with boldface fonts** and the replies with normal fonts, intended.

**Detailed replies to Anonymous Referee #1**

**Section 2.1. I believe a map of Nanjing and surrounding area, including its location relative to other cities in China would be helpful in understanding the site, and also in the analysis of the back trajectories and wind roses.**

Yes, we accept the reviewer's advice. We added a map of YRD as well as eastern part of China as figure 2b in the revised manuscript. This map represents the location of major cities as well as $\sigma_{sp}$ and SSA reported in this and previous studies. In trajectory and retroplume analysis, name and location of major cities are also marked in the plots.

**Page 5. How were the data points that were impacted by the faulty heater in the nephelometer identified? Also, what tests were done to ensure that this correction is not creating an artefact? For example, were those points across the monitoring period or in specific time period? How sensitive are your results to this correction (e.g. both the fact that it had to be done and the constants c and g that were used)? I understand that this correction was done in order to use the data, and that makes sense. However, I would recommend adding some more information on the impact of this assumption on the results.**

Data points that were affected by the faulty heater in the nephelometer were identified by $RH_{sample}$ recorded by the nephelometer's internal sensor. Data with $RH_{sample}$ >50% are regarded as affected by the faulty heater.

Most of these data were observed in the summertime when the ambient temperature was high. More specifically, the overwhelming majority of data in July and August, the majority of data in June and a small fraction of data in May, September, and October belong to the 'impacted data'. The scattering coefficients in these periods were corrected as explained in the text.

A simple comparison between corrected and uncorrected data is presented in the supplement (Figure S1). Two parameterizations were compared, the one presented by Carrico et al. (2003) and the other by Zhang et al. (2015). Please note that in the revised paper, c and g are taken from Carrico et al. (2003). We did this change for 2 reasons: on one hand, Carrico's paper also includes formulas for correcting backscattering coefficient, which is essential for correcting backscatter fraction. On the other hand, Zhang et al. (2013) has a shorter experiment period and states that their result is quite different from other studies so we choose not to use their results.

As we are not able to ensure that this correction is not creating any artifacts, we also used Mie theory to calculate optical properties based on the particle number size distribution measured with the DMPS. A detailed comparison between measured and modeled AOPs is presented in the supplement (e.g., Figures S2). The seasonal variations of AOPs are also in agreement with the theoretical calculation (Figure S4).

**Page 5 line 34, how was it decided to use the wavelength insensitive $C_{ref}$? How was it determined that this was the most appropriate?**

There is no unambiguos reasoning for using either a non-wavelength-dependent or a wavelength-dependent $C_{ref}$. Both are used in the literature. In the widely referenced paper on an intercomparison of aerosol absorption photometers (Müller et al., 2011) the authors used a non-wavelength-dependent $C_{ref}$ and so did Zotter et al. (2017) in a paper that presents a model for estimating the contributions of traffic and wood burning to BC by using AAE values. To keep the results somewhat more comparable the same approach was selected.

**Page 7 line 26, the Wang et al. reference for using a density of 1.7 g/cm$^{-3}$ is not local to this monitoring station. How was it determined that it is applicable to this dataset?**

We added the following text to the paper:

"The PM$_1$ concentrations were calculated by using the density of 1.7 g cm$^{-3}$ even though it was not determined with any physical measurements so an explanation is needed. The densitity of the major inorganic aerosol compounds ammonium sulfate and ammonium nitrate are 1.76 g cm$^{-3}$ and 1.725 g cm$^{-3}$, respectively (e.g. Tang, 1996).The density of sulfuric acid-coated soot has been estimated to be 1.7 g cm$^{-3}$ (Zhang et al., 2008). Ambient aerosol particles contain also many unknown compounds such as organics and also some water even at RH < 50 %. Densities of real atmospheric aerosols have been published from several regions. Just to name some, the mean apparent particle density of 1.6 ± 0.5 g cm$^{-3}$ was determined for urban aerosol by Pitz et al. (2003) and at a boreal forest site the average density was 1.66 ± 0.13 g cm$^{-3}$ (Saarikoski et al., 2005). Based on the above, it is reasonable to use the density of 1.7 g cm$^{-3}$ for the estimation of aerosol mass concentration from the number size distributions. It is clear, however, that this value is uncertain and that also in reality not constant as the chemical composition of aerosol varies."

Pitz, M., Cyrys, J., Karg, E., Wiedensohler, A., Wichmann, H.E. and Heinrich, J.: Variability of apparent particle density of an urban aerosol. Environmental Science & Technology, 37(19),4336-4342, 2003.

Saarikoski, S., Mäkelä, T., Hillamo, R., Aalto, P., Kerminen, V.-M., and, Kulmala, M.: Physico-chemical characterization and mass closure of size-segregated atmospheric aerosols in Hyytiälä, Finland, Boreal Environ. Res., 10, 385–400, 2005.

Tang, I.: Chemical and size effects of hygroscopic aerosols on light scattering aerosols, JGR 101, 19245 – 19250, 1996.

**Page 8 line 6, the first sentence in the first paragraph in 2.4 is not complete**

We agree, this sentence was poorly organized and difficult to understand and rewrote it as

"The NO$_x$ and NO$_y$ concentrations can be used as a semi-quantitative indicator of the age of air masses, i.e., oxidation time passed since the emission of NO$_x$ from its sources, mainly traffic."

**Page 8, line 10, I would recommend defining what is considered a "small" and "high" value.**

Thank you for your suggestion. "small" is a typing error, the usage of "high" is also not very clear in the original manuscript. We now define PA=0 as fresh and PA=1 as 'very aged.' Changes are highlighted in the revised manulscript.

**Page 10, paragraph starting on line 25 details the AAE range. It is stated here that because the wavelength independent Cref was used, these values are smaller than if a wavelength dependent Cref was used. In the conclusions, it is acknowledged that using a different Cref would lead to the conclusion that there is brown carbon. As this is an uncertainty that has an impact on the conclusion, I would recommend discussing this on page 10 as well.**

We agree and therefore added the following text to the paper.

It is simple to show that if the $C_{ref}$ depends on wavelength as $C_{ref,\lambda} = C_{ref,0}(\lambda/\lambda_0)^\alpha$ then $AAE(C_{ref,\lambda}) = AAE(C_{ref,0}) + \alpha$ where $AAE(C_{ref,0})$ is the AAE calculated from $\sigma_{ap}$ that were calculated from (4) using a constant $C_{ref} = C_{ref,0}$. The $C_{ref}$ of Arnott et al. (2005) has a wavelength dependency of $C_{ref,\lambda} \approx C_{ref,520}(\lambda/520 \text{ nm})^{0.181}$ which leads to AAE values of $AAE(C_{ref,\lambda}) \approx AAE(C_{ref,0}) + 0.181$ so that the average AAE would be 1.24 instead of 1.06 shown in Table 1. The above reasoning is an approximation only because the scattering corrections in (4) were not taken into account. These corrections also affect the AAE but an evaluation of them is out of the scope of the present paper. However, the AAE uncertainty further affects estimates of the sources of BC. The AAE has recently been used for estimating the contributions of burning fossil fuel and wood to BC. Zotter et al. (2017) used aethalometer data with a constant $C_{ref}$ and found that the AAE calculated from the wavelength pair 370 nm and 950 nm (AAE(370/950) was ~0.9 and ~2.09 for BC from traffic and wood burning emissions, respectively, by analyzing data collected in Switzerland. For the wavelength pair 470/950 the respective AAEs were ~0.9 and ~1.68 for BC from traffic and wood burning. By using these values and the average value of AAE(370/950) = 1.06, the model by Zotter et al. (2017) yields an estimate of ~8% of BC originate from wood burning and 92% from traffic. By using the wavelength pair 470/950 the average AAE of 1.04 the same model yields also ~8% contribution by wood burning. If a wavelength-dependent $C_{ref}$ were used that yields AAE values larger by 0.18, as discussed above, the average AAE(370/950) would be ~1.24 and the fraction of BC from wood burning ~18%. Zhuang et al. (2017) used a wavelength-dependent $C_{ref}$ for analyzing aethalometer data collected in the center of Nanjing and obtained the average AAE = 1.58 for the wavelength pair 470 nm and 660 nm. By using the same values and model this AAE would yield an estimated contribution of ~43% by wood burning. The difference of these results suggests that further, harmonized studies should be conducted.

**Page 11, section 3.2. In the reasons for the seasonal variation there is no mention of the possibility of a seasonal dependence of emissions impacting on the seasonal cycle. Why is that?**

Thanks for your suggestion. In the unrevised paper, we thought there should be no significant variations in emissions around SORPES and didn't discuss the potential effects, which is not appropriate. Transport of pollutants makes the strong seasonality of emissions able to influence AOPs at SORPES. The influence is not straightforward. Thus we added one individual section about the potential impact of seasonal cycles in the supplement section S5 and added a summary of the discussion in section 3.2.

**Page 12, line 8-10. The increase in SSA is attributed to new particle formation (NPF). However, transport of scattering particles (which is seen in the later analysis) could also lead to a peak in the afternoon when the aerosols are well-mixed. It is not clear why this was only attributed to new particle formation.**

There is a slight misunderstanding. It was written on P12 L8-10 that

" S*econdary aerosols formed by gas-to-particle conversion processes, for instance sulfates and nitrates were a likely cause for such a high SSA*".

This may indeed be interpreted so that we would claim that the particles formed by NPF the same morning would make the SSA grow. However, gas-to-particle conversion includes also condensational growth on existing particles and we also agree that the SSA increase cannot be attributed to NPF in the same day. Later in the ACPD paper P12, line 35-36 and P13, L1-3 it was already written

" *NPF produces small particles that are initially too small to affect the total light scattering. However, at the time of the NPF some of the older, larger particles still remained which resulted in a bimodal size distribution with a fast growing nucleation mode and an Aitken mode in the particle diameter range of about 70–90 nm. The species that condense on the newly-formed particles are typically light scattering inorganic and organic species and they also condense on the Aitken-mode particles.*"

But it is true, we did not mention transport and vertical mixing. In the revised version this is written more clearly. The change is that at the first instance we removed the text originally on P12 L8-10 and there simply describe how the SSA diurnal cycle looks like and leave the possible explanations to the subsection. The sentence ending on P13, L4 remains the same and the additional text is highlighted by yellow:

" ... At about 12:00 – 14:00 the newly-formed particles had grown into size range of about 20 – 50 nm, which has still a very small scattering cross section ($C_s = Q_s(\pi/4)D_p^2$, where $Q_s$ is the scattering efficiency) compared with that of the larger mode that at this time had grown by condensation of light-scattering species to ~100 nm or larger. This mode was probably only partially responsible for the high SSA. At the same time the boundary layer height (not shown) and the photochemical age reached the diurnal maximum (Figure 4f). This suggests that vertical transport of aged aerosol from upper levels contributes to the high SSA when the vertical mixing is at its strongest. It is also likely that the aged aerosol has a mixture of accumulation-mode particles transported from further away and more local aerosol that were formed and grown during the previous day and remained in a residual layer. Even though the above discussion

shows that particles formed by NPF cannot be responsible for the growth of SSA during the same day it is worth noting that NPF and SSA diurnal cycles do have a connection in our data. During the clearest NPF days, classified into the class I and II NPF according to the classification used by Qi et al. (2015), also the SSA diurnal cycles were the clearest and largest (Figure 5b and 5c). On the average new particle formation took place around 10 after which particles grew and reached optically significant sizes only late in the evening when SSA reached the minimum. By using the above reasoning this suggests that both the condensation on existing particles and the vertical mixing were then the strongest."

**This assumption that NPF events are responsible for this then also plays a role in the assessment that the aerosols responsible for higher scattering originated outside of Nanjing (page 15).**

There may be some misunderstanding here. The estimation shows that NPF and subsequent growth from within the boundaries of Nanjing cannot explain the high scattering coefficients. That is still true. If they were formed by NPF within Nanjing and subsequent growth within Nanjing these particles wouldn't have the time to grow so much that they would result in such a high scattering coefficient so they come from somewhere else. To this point the reviewer probably agrees.

But the reviewer seems to want that it is written that NPF has no effect on optical properties. But that is not true. All atmospheric aerosol originate either as primary or secondary particles. If they are secondary particles formed by NPF there is no doubt that they can grow by condensation and coagulation to sizes that have high scattering coefficients, given high enough concentrations of condensables and enough time.

However, we change the sentence on P14,L36 – P15L1 where it was written

" .. *At the weakest winds from the SW the contribution of the city to the amount of scattering particles may thus have been significan*t "

to

"At the weakest winds from the SW the contribution of the city to the amount of scattering particles may thus have been observed even if particles were formed by NPF."

And it is now further emphasized that the estimation above was based on the assumption that particles are formed by NPF. The effect of primary emissions have not been forgotten, there is now a text

"However, in a city a large fraction of aerosols are primary particles, especially BC emitted from vehicles in the size range of ~60 ± 20 nm (e.g., Bond et al., 2013; Kulmala et al., 2016). At GR $\approx$ 9 nm h$^{-1}$ it would take ~5 ± 2 hours for them to grow to sizes > 100 nm. In these cases also particles emitted from within Nanjing will affect scattering but their SSA would be lower than the observed SSA. This was estimated by a core-shell Mie code (Wu and Wang, 1991) that yields SSA < 0.6 at $\lambda$ = 520 nm for BC particles with a core of 60 ± 20 nm coated with 40 nm of scattering material, for instance ammonium sulfate. At the weakest winds from the SW sector the

observed SSA at $\lambda$ = 520 nm was > 0.9, clearly higher than it would be if particles were emitted as BC particles within Nanjing and coated by condensation, supporting the interpretation that sources of the high $\sigma_{sp}$ during weak winds are not only within Nanjing."

**this caveat is not stated in conclusions when this point is made again (page 22, line 21-22).**

On p.22, lines 21-22 it is written:

" *By using the observed particle growth rates and local wind it could be estimated that the center of Nanjing is too close to be the most important contributor to light-scattering particles.*"

Ok, here the main point is the same as above and it will be corrected to

" By using the observed particle growth rates and local wind it could be estimated that the center of Nanjing is too close to be the most important contributor to light-scattering particles if they were formed by NPF and subsequent growth within Nanjing. Primary particles, such as BC emitted from traffic, do have time to grow to optically significant sizes also from within Nanjing during weak winds.."

**On page 18, line 7 it is stated that there was a previous study looking at the dominance of NPF at the site. This is a key point that should be moved further up in the text to provide support to the discussion of NRF starting on page 12.**

We agree with your suggestion and moved this sentence to page 12.

**Page 14, line 22, the dilution with higher wind speeds is attributed to vertical mixing specifically, but I would imagine that horizontal transport and increased dispersion with high wind speeds would also play a role.**

This is true, we made a mistake by suggesting that vertical mixing would be the only expanation. We added the sentence

"The effect of horizontal transport, increased dispersion and shorter residence time within emission areas may also contribute to the decrease of $\sigma_{sp}$ and $\sigma_{ap}$. We don't start speculating further on the possible explanations, it would require extensive modeling that is out of the scope of the present paper."

**Page 15, line 20-21, what sources of emissions are in direction where the fresh traffic emissions were measured? If there are no obvious roads there, then the assessment of these as fresh traffic emissions would be incorrect.**

Yes, there is one obvious road in that direction. However, the potential effects from that road seem not to be enough to explain the observations. Based on the analysis presented in the supplement, section "S2.3 Explanations for high $\sigma_{ap}$ from SSE" we believe the explanation is the nighttime heavy traffic on country dirt roads to several small factories and mines located within about 3 ~10 km to the SE of SORPES.

We removed the sentence

"*The wind direction does not point to the center of the city so it is not that obvious*

*where it comes from.*"

We added this sentence to the main text:

" An analysis using aerosol chemical composition and wind data (supplement S2.3) suggests that the main cause for the high $\sigma_{ap}$ during winds from the SSE sector is the nighttime heavy traffic on country dirt roads to several small factories and mines located within about 3 ~10 km to the SE of SORPES."

**Page 15, line 23-25, the WNW sector was said to have very few points (page 14, line 25-26), thus how robust is the assessment of the age that comes up with controversial results? It does look from Figure 5 that the winds come from this direction mostly in winter, can that help to understand the potential source of pollution (e.g. NO$_x$ not emitted with BC)?**

First of all, in the ACPD text on p. 15, lines 23-26 it was written

"*Photochemically fresh air with –log(NOx/NOy) $\leq$ 0.2 was observed also with winds from the same WNW sector as the highly scattering aerosol. These are controversial results since to grow particles to the size range that scatters light at high efficiency requires time. Further, in the air masses from this sector BC concentrations and thus $\sigma_{ap}$ were low which suggests the NO$_x$ observed did not come from sources that emit also high BC concentrations.* "

This is the text that the reviewer refers to. The original text was actually erroneous in one important point. We claimed there that

" *... in the air masses from this sector BC concentrations and thus $\sigma_{ap}$ were low...* "

This sentence was written due a careless analysis of the plot. A more careful inspection of the contour polar plots for $\sigma_{sp}$ and $\sigma_{ap}$ (Figure 6a and b) shows that in that sector actually both of them have local maxima at the same WS-WD combinations. New coloring of the contour plots made them much clearer.

But what still is true is that the photochemical age is not as high as it could be expected for such high scattering coefficients.

In order to present whether the assessment of this wind sector is robust or not, a set of polar plots of medians and 25th and 75th percentiles for $\sigma_{sp}$, $\sigma_{ap}$ (Fig. S7), NO$_x$ and –log(NO$_x$/NO$_y$) (Fig. S8) are presented in the supplement, section S7. NO$_x$ concentration is higher and –log(NO$_x$/NO$_y$) is lower in the WNW sector than the overall averages indicating a fresh emission source really in that direction. There is a highway is located to the west of SORPES. The road surface is in perfect condition and the majority of vehicles on that highway are cars, not heavy vehicles. There are generally no traffic jams on that highway. In this kind of a situation, the emissions from the highway are mostly fresh NO$_x$ and VOCs with not so high concentrations of BC and road dust.

Wind blows from this direction mostly in winter, we totally agree this is one important aspect. The high $\sigma_{sp}$ were observed in winter pollution episodes during which SSA grew during the evolution of the episodes, as is discussed in section 3.5. Now the lower-than-expected photochemical age can be explained by the flow of the aged and grown particles over the

highway and mixing with fresh NO$_x$ emissions.

We rewrote the text as

" A careful inspection of the contour polar plots for σ$_{sp}$ and σ$_{ap}$ (Figure 5a and b) shows that in the WNW sector actually both of them have local maxima at the same WS-WD combinations. In these local maxima SSA is high, ~0.95 or higher, suggesting that the aerosol is aged. On the other hand, photochemically fresh air with –log(NO$_x$/NO$_y$) ≤ 0.2 was observed also with winds from the same WNW sector as the highly scattering aerosol. These are apparently controversial results since to grow particles to the size range that scatters light at high efficiency requires time. Wind blows from this direction mostly in winter (Figure 5d). The high σ$_{sp}$ was observed in winter pollution episodes during which SSA grew during the evolution of the episodes, as will be discussed below. There is a highway is located to the west of SORPES so the lower-than-expected photochemical age can be explained by the flow of the aged and grown particles over the highway and mixing with fresh NO$_x$ emissions. "

**Technical corrections**
**Page 1 line 26, "in" is duplicated**
corrected

**Page 1 line 32 "of" is duplicated**
corrected

**Page 1 line 34, "b" is referred to but is not yet defined.**
corrected

**Page 2 line 26, as stated, there are many studies in these areas, and thus the ones listed are only some of the studies. I would recommend using "e.g." before the references to indicate this.**
corrected

**Page 2 line 35, what year was SORPES station started?**
It started in 2011, and we add the year to the manuscript also.

**Page 4 line 2, "hygroscopic growth is usually significant when RH increase above 50% RH" is this statement specific to this site or a general statement?**
This is a general statement. The references (Zhang et al. 2015; WMO, 2016) are already in the ACPD paper, here is one older but still a valid one: on p. 5 of the WMO/GAW Report 153 on Aerosol Measurement Procedures (Baltensperger et al., 2003), it is written

"*Due to the hygroscopic nature of atmospheric aerosols, size increases with increasing relative humidity (RH) with the greatest change occurring above 50% relative humidity.*"

Baltensperger, U., Barrie, L., Fröhlich, C., Gras, J., Jäger, H., Jennings, S. G., Li, S.-M., Ogren, J. A., Wiedensohler, A., Wehrli, C., and Wilson, J.: WMO/GAW Aerosol Measurement Procedures, Guidelines and Recommendations, WMO/GAW No. 153, 67 pp., 2003.

**Page 4, line 27, For readers who are not familiar with SORPES, I would recommend moving this reference for the overview of the site (Ding et al 2013 and 2016) at the beginning of this section.**

We totally agree with that, we add 'Supporting measurements were measured for the same period (Ding et al., 2013b and 2016b)' as the first sentence of this section

**Section 2.3.4, the terms for equations 9-11 are not all defined in the text.**

The definition is added and highlighted in the text.

**Page 8, line 34, PM2.5 are particles with an aerodynamic diameter less than or equal to 2.5 um.**

We changed to this more accurate definition.

**Page 12, line 5, "wee hours" is colloquial; I would recommend using another term.**

We changed "wee hours" to "Early morning".

**Detailed replies to Anonymous Referee #2**

**1) The heater in the nephelometer did not work for much of the 2-year observations period. Given the warm temperatures and high humidity during much of the year and the operation of the instruments in a (presumably cooler) indoor laboratory, there may be considerable aerosol water remaining even if the data are restricted to calculated RH<50%. The authors have not attempted to compare the SMPS size distribution measurements with the nephelometer by using Mie theory to calculate ambient scattering. There should be a short section that compares all measurements to whatever extent possible, including scattering, backscatter fraction, and aerosol mass. In other words, show internal consistency for the measurements analyzed. Then the reader would be more assured that the instruments are operating within their expected uncertainty.**

Thank you for the suggestion. Comparisons between measured scattering and backscattering coefficient with modeled ones by using the particle number size distributions and a Mie code are now presented in the supplement, section S2. There is also a comparison between humidity-corrected $\sigma_{sp}$ with $PM_{2.5}$ mass concentrations, supplement section S3.

**2) Measurement variability is often discussed as a mean and standard deviation (e.g., in the abstract). However, these metrics are not descriptive when the parameter is not normally distributed, as is the case for most of the extensive parameters. In time plots, the data are shown to vary on a logarithmic scale. Thus it would be more logical to calculate, display, and discuss geometric means and geometric standard deviations. I appreciate that the 10th and 90th percentile values are also used as an indicator of variability; this is good and should be retained.**

Thanks for the suggestions, we agree it is more logical also to present and discuss geometric means and geometric standard deviations for some external properties. Geometric means and geometric standard deviations of $\sigma_{sp}$, $\sigma_{ap}$ and $PM_{2.5}$ are presented in Table 1 and discussed in section 3.1 of the revised manuscript.

However, in the abstract and many other places, we still use averages and standard deviations. The main reason is that the majority of previous studies use them so it is easier for comparison with other studies if we present the value in the same way.

**3) The time plots (Figs 1 and 8) are not very useful because there are so many parameters displayed. Figures 1 and 8 in particular are extremely dense, and it is very hard to see systematic changes during the "episodes" that are discussed in the text. I recommend focusing on one exemplary "episode" for Figure 8 and eliminating the other times shown on this graph. Then average values, described in a table, could be shown to describe the remaining episodes, e.g. length of each episode, changes in optical properties from episode start to finish, etc. This would be more quantitative and useful to the reader. Please make a quantitative definition of an "episode" and use it consistently to identify these events.**

Thank you for the suggestions, We redesigned and plotted Figs 1 and 8.

Fig. 1: we simplified the figure. We only keep three parameters: $\sigma_{sp}$, SSA, and $PM_{2.5}$. The updated Figure 1 is aimed to present 1) the consistency of different parameters; 2) that both extensive and intensive AOPs change significantly when pollution level (use $PM_{2.5}$ as an indicator) changes; 3) the idea 'The extent of rapid change in multi-day scale is beyond the extend diurnal scale (range between 10th and 90th percentiles as the indicator) and seasonal scale' can be seen directly from time series.

Fig. 8: We made the presented period shorter and only keep two episodes. We chose to show two episodes instead of one episode to illustrate that episodes usually occur one right after another with very similar characteristics. In the revised manuscript the figure numbers changed due to the addition of a new Fig 5. so the new figure is Fig. 9. The original Fig. 8 was not deleted, we moved it to the supplement, Fig. S11. There are two reasons we still opted to keep it: 1) the backtrajectory clusters and the retroplumes of Figure 8 were calculated for the whole 4-month period, not just the shorter period so it is reasonable also the time series for the respective period is shown and 2) the full winter time series reveals even more clearly some of the main regularities of the evolution of AOPs during the growrh phase of the episodes and the abrupt changes at the end of the episodes.

An 'episode' is defined as a period that meets following criteria: 1) minimum three days; 2) PM2.5 concentrations at both start and end of the period are significantly lower than the maximal value during the episode; 3) only one sharp/flat peak during the whole period. The definition was added to the manuscript (section 3.1). It is worthwhile to note that objective definition is not easy to achieve and a semi-subjective definition is enough for brief discussion in this study.

**4) The authors may wish to consider explicitly if biomass burning (mostly agricultural) could explain the cases of very high scattering but low absorption coming from the west-northwest direction**

Biomass burning is not likely to not explain the observation. Wind blows from the WNW sector mostly in winter but the agricultural biomass burning season is not then. The potential effect of biomass burning as an explanation for high $\sigma_{sp}$ from WNW is analyzed more in the supplement section S7.2. Based on the chemical composition, aerosol from biomass burning could in principle have contributed some. The high $\sigma_{sp}$ were observed in winter pollution episodes during which SSA grew during the evolution of the episodes, as is discussed in section 3.5. Now the lower-than-expected photochemical age can be explained by the flow of the aged and grown particles over the highway and mixing with fresh $NO_x$ emissions.

**5) On p. 14, line 30, an assumption is made that new particle formation and growth to optically significant sizes are necessary to explain increases in aerosol scattering. In**

**fact, light scattering is roughly proportional to particle volume or mass because the scattering efficiency increases very approximately linearly across the broad accumulation mode, leaving scattering proportional to diameter cubed. Most of the increase in mass, hence light scattering, is almost certainly occurring on this accumulation mode. The growth of recently formed particles is very unlikely to compete with condensational growth on existing accumulation mode particles because of this strong diameter dependence of scattering.**

Reviewer 1 paid attention to the same point. It is true, particles formed by NPF cannot contribute significantly to scattering or the growth of SSA during the same day. In section 3.3 it is now written

" However, at the time of the NPF some of the older, larger particles still remained which resulted in a bimodal size distribution with a fast growing nucleation mode and an Aitken mode in the particle diameter range of about 70–90 nm (Figure 5). The species that condense on the newly-formed particles are typically light scattering inorganic and organic species and they condense also on the Aitken-mode particles. At about 12:00 – 14:00 the newly-formed particles had grown into the size range of about 20 – 50 nm, which has still a very small scattering cross section ($C_s = Q_s(\pi/4)D_p^2$, where $Q_s$ is the scattering efficiency) compared with that of the larger mode that at this time had grown by condensation of light-scattering species to ~100 nm or larger. This mode was probably only partially responsible for the high SSA. At the same time the boundary layer height (not shown) and the photochemical age reached the diurnal maximum (Figure 4f). This suggests that vertical transport of aged aerosol from upper levels contributes to the high SSA when the vertical mixing is at its strongest. It is also likely that the aged aerosol has a mixture of accumulation-mode particles transported from further away and more local aerosol that were formed and grown during the previous day and remained in a residual layer. Even though the above discussion shows that particles formed by NPF cannot be responsible for the growth of SSA during the same day it is worth noting that NPF and SSA diurnal cycles do have a connection in our data. During the clearest NPF days, classified into the class I and II NPF according to the classification used by Qi et al. (2015), also the SSA diurnal cycles were the clearest and largest (Figure 5b and 5c). On the average new particle formation took place around 10 after which particles grew and reached optically significant sizes only late in the evening when SSA reached the minimum. By using the above reasoning this suggests that both the condensation on existing particles and the vertical mixing were then the strongest.

**6) In Section 3.6.2 it is confusing to describe the radiative effect as "decreasing" or "getting smaller" as light scattering increases. The magnitude of the effect is increasing; the sign of the effect is negative. I recommend explicitly saying that the magnitude of the RFE is increasing and use this consistently throughout the text.**

We made a clarification in 3.6.2 and use 'magnitude of the RFE is increasing' to describe changes for example from -18 to -20 W m$^{-2}$ throughout the text.

**Technical comments:**

1) **P1, line 34. The parameter "b" is not defined.**

corrected

2) **P2, line 15. Is the acronym "LAC" ever used again? There is no need to define an acronym that is not used later.**

The acronym "LAC" was removed.

3) **P3, line 7. Change "month" to "months".**

Corrected

4) **P10, line 23. Change "there" to "their"**

Corrected

5) **P16, line 23. Correct "to1".**

Corrected

6) **Fig. 1. The choices of gray-scale error bands make the whole graph look out-of- focus. Is there some better way to do this?**

We changed the color of the median lines into red. It looks better. Another potential factor for such out-of-focus is that the values of 90[th] and 10[th] percentiles are too close to each other compared with the multi-day scale variation.

7) **Fig. 1. I don't see any "blue vertical lines". It's not at all clear how the vertical lines that are shown on the graph are related to any "episodes".**

We agree the "blue vertical lines" is not clear and even cause confusion, and we removed them.

8) **Fig. 1. The scales/units for the absorption coefficient are incorrect.**

Thank you for point that out, we corrected it

9) **Fig. 4. What is the color scale in Fig. 4f? Is this dN/dlog(Dp)? Is this an average (mean) sized distribution over the entire period of the observations? Is it rational to average size distributions in this way? For example, an accumulation-mode-dominated size distribution averaged with one dominated by an Aitken mode peak might result in a size distribution that represents neither case. Would it be better to show temporal evolution of modal parameters (e.g., accumulation mode Dgeom, Aitken/nuclei mode Dgeom) calculated and displayed as in Fig a-e?**

The color scale is dN/dlog(Dp), it is an averaged size distribution over the entire period. Some previous studies averaged the size distribution in this way (Qi et al., 2015). Such plot gives all overall information about how particle usually distributed during the day but not able to represents any certain type of size-distribution as the reviewer mentioned.

The reviewer suggests showing temporal evolution of modal parameters. We think this is a good suggestion; however, the size distribution in SORPES is usually very complex, several modes may exist at the same time thus makes mode fitting very tedious. Since the

modal structure of the size distributions is not the focus of the present paper we did not start this time-consuming task. Instead, Fig 4f was replaced by diurnal cycles of GMD and VMD. Both GMD and VMD can serve as representative of particle size, but they are sensitive to different size range. The diurnal cycle of the size distribution was moved to a new figure 5. As a conseqence, the numbering of the rest of the figures changed.

**10) Fig. 4. Can you show some sparsely plotted error bars that represent the 10[th] and 90[th] percentile values on the plots? This might make the graphs too crowded, but there needs to be some way to show if the changes in the parameters are significant relative to the hourly variability of the data.**

We agree, this is in principle a good idea. To avoid making the graphs too crowded, we plotted similar diurnal plot but for 10[th] 50[th] and 90[th] percentiles in the supplement Fig S6. The problem with that figure is, however, that the percentiles were calculated from the whole data whereas in Fig. 4 the diurnal cycles were calculated for each season separately. In Fig S6 there are now 6 subplots. If they were made separately for the 4 seasons, the number of plots would have increased to 24 which we considered to be unneccessarily many.

**11) Fig. 5. Please explain these color contour plots. What is a "counter polar plot". What do the colors represent?**

"counter polar plot" was a typing error, it should be "contour polar plots". The original Fig 5a~c, now Fig 6a-c are 'contour plots of AOPs in polar coordinates'. The colors show the arithmetic mean values of $\sigma_{sp}$, $\sigma_{ap}$ and $-\log(NO_x/NO_y)$.

**12) Fig. 6. Please explain the color scale and the units. What does "retroplume" as a value mean exactly?**

The unit is $m^{-3}$ hr. This explanation was now added to section 2.5:

In this study, a retroplume refers to the concentration (in $m^{-3}$) averaged between 0 m and 100 m and integrated for 1 hour. So the unit is $m^{-3}$ hr. In each run, HYSPLIT was set to model the release of 3000 particles each hour which the model treated as a release of 1 particle every 1.2 seconds at SORPES, and track the backward trajectory of each one. The 'concentration' indicates the location of all those particles at a given moment. The 'retroplume' as a integrated concentration, however, also gives information on the residence time of the particles within a certain grid cell.

**13) In Fig. 7 the trajectories are labeled "1", "2", etc. In Fig. 8 they are given names. Please use the names instead of numbers in Fig. 7.**

Names have already added on Fig. 7

**14) The color scale for Fig. 12 is readable by most color-blind people, while the color scales on the other figures are not. Please use this color scale on all figures for the 7% of readers who are color-blind.**

We changed all the color scale accordingly.

**15) Table 1. Show geometric means and standard deviations.**

We added the geometric means and standard deviations in table 1.

**16) Table 2. Add the current results to this table so we can compare directly.**

We added the current results in table 2.